# Unsupervised Object Detection Pretraining with Joint Object Priors Generation and Detector Learning

**Yizhou Wang**[1,3]*    **Meilin Chen**[1,3]*    **Shixiang Tang**[2] †    **Feng Zhu**[3]    **Haiyang Yang**[5]

**Lei Bai**[4]    **Rui Zhao**[3,6]    **Yunfeng Yan**[1]    **Donglian Qi**[1]    **Wanli Ouyang**[4,2]

[1]Zhejiang University    [2]The University of Sydney    [3]SenseTime Research

[4]Shanghai AI Laboratory    [5]Nanjing University

[6]Qing Yuan Research Institute, Shanghai Jiao Tong University, Shanghai, China

`{yizhouwang, merlinis, yvonnech, qidl}@zju.edu.cn`   `stan3906@uni.sydney.edu.au`

`{zhufeng, zhaorui}@sensetime.com`   `baisanshi@gmail.com`

`hyyang@smail.nju.edu.cn`   `wanli.ouyang@sydney.edu.au`

## Abstract

Unsupervised pretraining methods for object detection aim to learn object discrimination and localization ability from large amounts of images. Typically, recent works design pretext tasks that supervise the detector to predict the defined object priors. They normally leverage heuristic methods to produce object priors, *e.g.,* selective search, which separates the prior generation and detector learning and leads to sub-optimal solutions. In this work, we propose a novel object detection pretraining framework that could generate object priors and learn detectors jointly by generating accurate object priors from the model itself. Specifically, region priors are extracted by attention maps from the encoder, which highlights foregrounds. Instance priors are the selected high-quality output bounding boxes of the detection decoder. By assuming objects as instances in the foreground, we can generate object priors with both region and instance priors. Moreover, our object priors are jointly refined along with the detector optimization. With better object priors as supervision, the model could achieve better detection capability, which in turn promotes the object priors generation. Our method improves the competitive approaches by **+1.3 AP**, **+1.7 AP** in 1% and 10% COCO low-data regimes object detection.

## 1 Introduction

Object detection is a fundamental task in computer vision, whose goal is to predict a set of bounding boxes and category labels for all objects of interest in the images. The success of modern detectors [4, 35, 22, 14, 27, 29, 31, 32, 41, 42] relies on the large-scale datasets with precious annotations which are very costly and even infeasible for enormous images available in the Internet. Therefore, unsupervised pretraining methods [12, 3] for object detection are proposed to ease the burden of these human annotations, which enable the detectors to be fast deployed in the real world by finetuning the pretrained model with only a few annotated images.

While various unsupervised pretraining methods, *e.g.,* SwAW [5], BYOL [18], DetCo [51], have been shown to learn good backbones for object detection [28, 52, 46, 37], only a few methods are able to pretrain the detection head, which is also a key component of a complete detection architecture [24, 20, 25]. To enable the localization of the pretrained detection head, a few recent

---

*Equal contribution. The work was done during an internship at SenseTime.

†Corresponding author.

36th Conference on Neural Information Processing Systems (NeurIPS 2022).

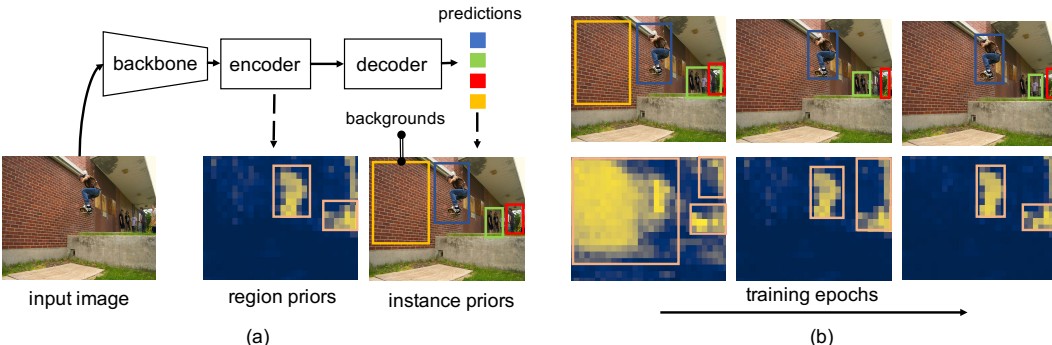

Figure 1: (a) JoinDet extracts supervision from the knowledge learned by the model itself. The self-attention highlights foregrounds and the detector outputs instance predictions. (b) During training, object prior generation and detector optimization can mutually guide each other, leading to better region priors and better instance priors.

methods use hand-craft methods [58, 45, 11, 1] to generate object priors on unlabeled images to construct the pretext tasks. Specifically, UP-DETR [12] and DETReg [3] generate object priors by randomly cropping patches and selective search [45], respectively. Then, both of them require the detection model to learn to detect those object priors for pretraining. Although these methods bring possibilities of pretraining detection heads, we argue that they suffer from two main drawbacks: (1) Both random cropping and selective search can only provide limited object-related supervision. For random cropping, it does not explicitly focus on objects. Selective search, as a hand-crafted and heuristic box generation method, relies on low-level features of input images to generate object priors, which is an extra time-consuming non-deep-learning process. (2) The supervision can not be progressively refined during training. For example, the object priors of selective search are fixed through the pretraining process, which limits the upper bound of the methods.

In this paper, we propose a novel unsupervised object detection pretraining method that can generate object priors and learn the detector jointly. In our method, we mine the supervision (object priors) based on the model itself, and show that the generated object priors can be jointly refined with detection optimization. Inspired by DINO [6], the self-attention maps can generate region prior bounding boxes, as they highlight the foreground regions (see Fig. 1(a)). Meanwhile, the output bounding boxes of the detector can generate instance prior bounding boxes, although may be located in the backgrounds (see Fig. 1(a)). By combining knowledge in region priors and instance priors together, we can generate reliable object priors only from the detector model. Furthermore, the object prior generation and detector optimization can be mutually evolved (see Fig. 1(b)). Supervised by reliable object priors, the model can gradually learn better object localization information, producing better region and instance priors for better object priors.

Given the object priors generated along with detector optimization, the supervision signal for the detector learning shifts quickly at the beginning, which leads to undesirable learning instabilities. In order to stabilize the pretraining process, both preceding and current object priors should be considered, and the update should be implemented in a slowly progressing way. To this end, we accomplish this regularization by designing a momentum box update strategy. Specifically, we match the current object priors to their neighboring and proceeding priors, and implement the update as a momentum-based average of their object prior locations.

Our contributions are three-fold. (1) We propose to mine supervision from the knowledge of the model to be learned itself for unsupervised object detection pretraining, which can generate supervision signals. (2) We design an unsupervised object detection pretraining method with **Joi**nt regio**n** priors generation and **Det**ector learning (JoinDet) which jointly generates reliable object priors and learns object detection in a progressive way. (3) We propose a Box Smooth Module to stabilize the pretraining process on JoinDet. Our method outperforms state-of-the-arts on low data-regime object detection on MS COCO, and full-data finetuning on PASCAL VOC. Detailed related works are provided in supplementary materials.

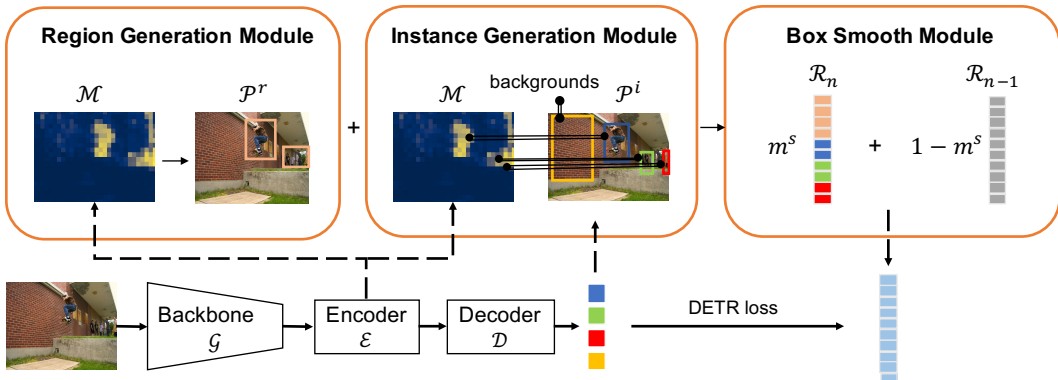

Figure 2: JoinDet generates region priors from the transformer encoder and instance priors from the transformer decoder. Reliable object priors can be achieved by concatenating region and instance priors. The Box Smooth Module then smooths object priors refinement in a momentum manner to preserve previous more accurate supervision and stabilize the pretraining process.

## 2 Methodology

Our goal is to mine effective supervision from the knowledge learned by the model itself for pre-training without annotations. To this effect, we propose an unsupervised object detection pretraining method with joint object priors generation and detector learning (JoinDet) which dynamically generates reliable object priors and learns object detection. Specifically, we first extract the eigen attention map from the transformer to highlight foregrounds. We then generate region priors from the eigen attention map using the Region Generation Module and choose reliable predictions as instance priors using the Instance Generation Module. By combining region and instance priors, reliable object priors are generated and set as the pseudo-labels for the object detection pretraining process. To stabilize the optimization process, we further propose a Box Smooth Module to smooth the refinement of object priors during pretraining.

During pretraining, the supervision is alternated between two steps: (1) generating reliable object priors by combining region priors and instance priors, including Step 1-3 below. (2) optimizing the detection model with reliable object priors (Step 4). Specifically,

*Step1*: *Generate region priors (Sec. 2.1).* At $n$-th epoch, given an image $\mathbf{x}$, the output patch features $\mathcal{F}$ of the transformer encoder $\mathcal{E}$ can be computed by $\mathcal{F} = \mathcal{E}(\mathcal{G}(\mathbf{x}))$, where $\mathcal{G}$ is the backbone network. To better highlight foregrounds, we generate the eigen attention map $\mathcal{M}$ with $\mathcal{M} = \mathcal{K}(\mathcal{F})$, which is detailed later. The region priors $\mathcal{P}^r$ can then be generated by the **Region Generation Module**.

*Step2*: *Generate instance priors (Sec. 2.2).* With the DETR-based detection frameworks, object queries $\mathbf{q}$ and patch features $\mathcal{F}$ are fed into a transformer decoder $\mathcal{D}$ together to generate foreground prediction boxes, *i.e.*, $\mathcal{P}_n = \mathcal{D}(\mathcal{F}, \mathbf{q})$. Guided by the eigen attention map $\mathcal{M}$, instance priors can be generated by the **Instance Generation Module** denoted as $IG$, i.e., $\mathcal{P}_n^i = IG(\mathcal{P}_n, \mathcal{M})$.

*Step3*: *Smooth object priors refinement by the Box Smooth Module (Sec. 2.3).* Reliable object priors $\mathcal{R}_n$ can be generated by combining region priors $\mathcal{P}_n^r$ and instance priors $\mathcal{P}_n^i$ with $\mathcal{R}_n = \mathcal{P}_n^r \oplus \mathcal{P}_n^i$. Given $\mathcal{R}_n$ and the last object priors $\mathcal{R}_{n-1}$, the momentum averaged object priors are computed by $\mathcal{R}_n \leftarrow BS(\mathcal{R}_n, \mathcal{R}_{n-1})$, where $BS$ represents the **Box Smooth Module**. The $\mathcal{R}_0$ is initialized with the region priors $\mathcal{P}_0^r$ before pretraining the detector.

*Step4*: *Train the model with updated supervision using set prediction loss (Sec. 2.4).* Given model predictions and updated supervision $\mathcal{R}_n$, we train the transformer parameters using set prediction loss.

### 2.1 Region priors generation

There are two steps of region priors generation. First, compute the eigen attention map on patch features from the transformer encoder. Second, extract the foreground bounding boxes as region priors.

**Generate the eigen attention map.** Our main motivation follows [47, 39] is that the eigen attention maps in the vision transformer can well highlight salient foreground objects. Specifically, given an input image $\mathbf{x}$ and DETR with a backbone $\mathcal{G}$, a transformer encoder $\mathcal{E}$, the eigen attention map $\mathcal{M}$ is extracted from the transformer encoder by

$$\mathcal{M} = \mathcal{K}(\mathcal{F}) = \mathcal{K}(\mathcal{E}(\mathcal{G}(\mathbf{x}))), \tag{1}$$

where $\mathcal{K}$ represents the computation method of the eigen attention map mentioned in [47], and $\mathcal{F} \in \mathbb{R}^{h \times w}$ represents output patch features from transformer encoder. The details of the eigen attention map computation method $\mathcal{K}$ are elaborated in supplementary materials.

**Extract region priors.** As foregrounds are highlighted in the eigen attention map, the bounding boxes of foreground regions can be extracted as region priors using Region Generation Module $RG$. Concretely, $\mathcal{M}$ is firstly partitioned into background parts (denoted by 0) and foreground parts (denoted by 1) by a bi-partition mask $\mathbf{m}$:

$$\mathbf{m}_{ij} = \begin{cases} 1, & \text{if } \mathcal{M}_{ij} > \alpha, \\ 0, & \text{otherwise,} \end{cases} \tag{2}$$

where $i, j$ represent the position index, $\alpha = \frac{1}{hw} \sum_{i=1}^{h} \sum_{j=1}^{w} \mathcal{M}_{ij}$ is the average score of $\mathcal{M}$. Region priors $\mathcal{P}^r$ are then generated by computing the minimum bounding rectangles of all connected foreground regions.

In addition, at the beginning of pretraining, to generate the initial supervision $\mathcal{R}_0$, we compute $\mathcal{P}_0^r$ using the unsupervised pretrained backbone $\mathcal{G}$ by

$$\mathcal{R}_0 = \mathcal{P}_0^r = RG(\mathcal{K}(\mathcal{G}(\mathbf{x}))). \tag{3}$$

Although region priors represent highlighted foregrounds in the eigen attention map, they do not divide foregrounds into foreground instances, and therefore instance priors should be mined.

## 2.2 Instance priors generation

The foreground predictions from the box prediction branch of DETR provide instance information. With object queries $\mathbf{q}$ and output patch features $\mathcal{F}$ passing to transformer decoder $\mathcal{D}$, DETR generates a set of box predictions $\mathcal{P}$ by

$$\begin{aligned} \mathcal{P} &= \mathcal{D}(\mathcal{F}, \mathbf{q}) \\ &= \{\mathcal{P}_j = (\hat{\mathbf{b}}_j, \hat{p}_j)\}_{j=1}^{N}, \end{aligned} \tag{4}$$

where $N$ denotes the number of output predictions and $\hat{\mathbf{b}}_j, \hat{p}_j$ denotes the predicted bounding box, and logits in foreground-background predictions, respectively. The prediction score $\hat{s}_j$ can be computed by $\hat{s}_j = Softmax(\hat{p}_j)$. As the model only predicts background (class label 0) and foreground (class label 1), we only reserve foreground predictions $\mathcal{P}^f$.

$$\mathcal{P}^f = \{(\hat{\mathbf{b}}_j, \hat{s}_j) | \arg\max \hat{p}_j = 1\}_{j=1}^{N}. \tag{5}$$

Although foreground predictions can define instances, some of them might correspond to the background, which should be removed by the Instance Generation Module to improve reliability.

**Instance Generation Module** $IG$, in which foreground predictions $\mathcal{P}^f$ and the eigen attention map $\mathcal{M}$ interact together to eliminate background predictions and generate instance priors $\mathcal{P}^i$. Specifically, boxes with low prediction scores in $\mathcal{P}^f$ are removed with

$$\mathcal{P}^i = IG(\mathcal{P}^f, \mathcal{M}) = \{\mathcal{P}_j^f | \mathcal{P}_j^f \in \mathcal{P}^f, j \in [1, N], N_f/N_t \geq t\}, \tag{6}$$

where $\mathcal{P}_j^f$ is the element in $\mathcal{P}^f$, $N$ is the number of elements in the $\mathcal{P}^f$, $N_f$ is the number of foreground pixels in the eigen attention map $\mathcal{M}$ bounded by $\mathcal{P}_j^f$, $N_t$ is the number of total pixels in $\mathcal{M}$ bounded by $\mathcal{P}_j^f$. Here, $t$ is a threshold set to 0.5. Guided by highlighted foregrounds in $\mathcal{M}$, background predictions in $\mathcal{P}^f$ can be removed. The remaining predictions are treated as reliable instance priors $\mathcal{P}^i$.

## 2.3 Box Smooth Module

Given generated region priors $\mathcal{P}^r$ and instance priors $\mathcal{P}^i$, reliable object priors $\mathcal{R}$ can therefore be generated by

$$\mathcal{R} = \mathcal{P}^r \oplus \mathcal{P}^i, \tag{7}$$

where $\oplus$ denotes the concatenation operation. We use reliable object priors $\mathcal{R}$ to supervise the pretraining process.

As region priors $\mathcal{P}^r$ and instance priors $\mathcal{P}^i$ are dynamically updated along with network optimization, the fast evolution of reliable object priors $\mathcal{R}$ may lead to the instability of the training process. Motivated by Weight Box Fusion [40], we propose a Box Smooth Module to overcome these problems, which preserves accurate object priors from previous supervision and reduces the perturbation using a momentum update strategy. Specifically, given current object priors $\mathcal{R}_n$, together with previous supervision $\mathcal{R}_{n-1}$, we cluster all boxes in $\mathcal{R}_n$ and $\mathcal{R}_{n-1}$ using IoU metric to generate $K$ box clusters. We recompute the box coordinates and scores using all boxes in the $k$-th cluster to generate a new box $\mathbf{b}_k$. The supervision $\mathcal{R}_n = \{(\mathbf{b}_k, s_k)\}_{k=1}^K$ is then updated in a momentum strategy by

$$\mathbf{b}_k = \frac{\sum_{i=1, \mathbf{b}_i \in \mathcal{B}}^T m^s s_i \mathbf{b}_i + \sum_{j=1, \mathbf{b}_j \in \mathcal{R}}^U (1-m^s) s_j \mathbf{b}_j}{\sum_{i=1}^T m^s s_i + \sum_{j=1}^U (1-m^s) s_j}, s_k = \frac{\sum_{i=1}^T s_i + \sum_{i=1}^U s_j}{T+U}, \tag{8}$$

where $k$-th cluster contains $T$ boxes in $\mathcal{R}_{n-1}$ and $U$ boxes in $\mathcal{R}_n$, and $m^s \in [0,1]$ represents the momentum coefficient to adjust the influence of previous supervision $\mathcal{R}_{n-1}$ and current object priors $\mathcal{R}_n$. $m^s$ is empirically set as 0.45. When $m^s$ is set to 1, the supervision will not be updated. When $m^s$ is set to 0, the supervision will be directly changed to current object priors $\mathcal{R}_n$. When initializing the pretraining ($n = 0$), we set $\mathcal{R}_0 = \mathcal{P}_0^r$ as the initial supervision. With such a design, we smooth the refinement of object priors to stabilize the training process.

**Discussion.** As no labels are used to supervise the model, generated region priors and instance priors have relatively large variations between training epochs. The Box Smooth Module is a critical design in the framework, which suppresses inaccurate previous supervision and slows down the shifting speed of supervision to guarantee stable optimization of the model. Note that Weighted Box Smooth is just a simple implementation to refine the supervision in a momentum update strategy, which is different from its original use to ensemble boxes from different object detection models.

## 2.4 Objective function

Here, we describe the objective function of JoinDet during pretraining. Following DETReg [3], JoinDet has three prediction heads: $f_{cls}$ which predicts the class (foreground or background), $f_{box}$ which predicts bounding boxes, and $f_{emb}$ which reconstructs features of the predicted regions. Assume $K$ boxes $\{(\mathbf{b}_k)\}_{k=1}^K$ left after the Box Smooth Module, use a pretrained SWAV [5] model to extract features $\mathcal{Z} = \{\mathbf{z}_k\}_{k=1}^K$ of box regions, assign class label $c_k = 1$, and let targets $y = \{y_k = (c_k, \mathbf{b}_k, \mathbf{z}_k)\}_{k=1}^K$. Let $\{\mathbf{v}_j\}_{j=1}^N$ denote $N$ query embedding outputted by transformer decoder, the JoinDet outputs can be denoted as: $\hat{p}_j = f_{cls}(\mathbf{v}_j), \hat{\mathbf{b}}_j = f_{box}(\mathbf{v}_j), \hat{\mathbf{z}}_j = f_{emb}(\mathbf{v}_j)$. The model predictions can be defined as $\hat{y} = \{\hat{y}_j = (\hat{p}_j, \hat{\mathbf{b}}_j, \hat{\mathbf{z}}_j)\}_{j=1}^N$.

Following DETR [4], we use Hungarian bipartite matching algorithm [26] to match $y_j$ and $\hat{y}_j$ by finding the permutation $\sigma$ that minimizes the optimal matching cost between $y_j$ and $\hat{y}_j$:

$$\sigma = \underset{\sigma \in \sum_N}{\arg\min} \sum_j^N \mathcal{L}_{match}(y_j, \hat{y}_{\sigma(j)}), \tag{9}$$

where $\mathcal{L}_{match}$ is the pairwise matching cost matrix. After finding the optimal $\sigma$, the set prediction loss is defined as:

$$\mathcal{L}_{Hungarian}(y_j, \hat{y}_j) = \sum_{j=1}^N [\lambda_c \mathcal{L}_{cls}(c_j, \hat{p}_{\hat{\sigma}(j)}) + \mathbb{1}_{c_i \neq \emptyset} \lambda_b L_{box}(\mathbf{b}_j, \hat{\mathbf{b}}_{\hat{\sigma}(j)}) + \lambda_e \mathcal{L}_{emb}(\mathbf{z}_j, \hat{\mathbf{z}}_{\hat{\sigma}(j)})],$$
$$\tag{10}$$

where $\mathcal{L}_{cls}$ is the classification loss which can be implemented via the cross-entropy loss or Focal Loss, $\mathcal{L}_{box}$ is based on $L_1$ loss and GIoU Loss [36]. Following DETReg, we set $\mathcal{L}_{emb}$ as $\mathcal{L}_{emb}(\mathbf{z}_j, \hat{\mathbf{z}}_{\hat{\sigma}(j)}) = \|\mathbf{z}_j - \hat{\mathbf{z}}_{\hat{\sigma}(j)}\|_1$.

Table 1: Low-Data regimes object detection results. Finetuning using 1% and 10% of the labels data on COCO. Following DETReg [3], all models are pretrained for 50 epochs on COCO without labels and finetuned using 1% and 10% labeled data. Results on $val2017$ are reported.

| Pretraining | COCO 1% | COCO 10% |
|---|---|---|
| Supervised | 11.31±0.3 | 26.34±0.1 |
| SwAV [5] | 11.79±0.3 | 27.79±0.2 |
| ReSim [50] | 11.07±0.4 | 26.56±0.3 |
| DETReg [3] | 14.58±0.3 | 29.12±0.2 |
| JoinDet | **15.89±0.2** | **30.87±0.1** |

## 3 Experiment

### 3.1 Implementation details

**Architecture.** We choose Deformable-DETR [57] as the baseline model for its high accuracy and fast convergence. Following DETReg [3], in prediction heads, $f_{box}$ and $f_{emb}$ are multi-layer perceptrons (MLPs) with 2 hidden layers of size 256 followed by a ReLU [34] layer. The output dimensions of $f_{box}$ and $f_{emb}$ are 4 and 256, respectively. We use a single fully-connect layer with output dimension 2 as $f_{cls}$.

**Datasets.** For a fair comparison, we pretrain our JoinDet on ImageNet-1K [13], ImageNet-100 and MS COCO [30] without labels, and evaluate our method on MS COCO and PASCAL VOC. ImageNet-1K contains 1.28M object-centric images with 1000 class categories. Similar to other works [38, 21, 56], we use a subset of ImageNet-1K that contains 125K images and 100 classes as ImageNet-100. MS COCO is a popular object detection dataset that contains 121K scene images with 80 object categories labeled with bounding boxes. PASCAL VOC contains 20K images with 21 object categories.

**Training.** Following DETReg, we initialize the ResNet50 [23] backbone of JoinDet with SwAV [5], which is fixed throughout the training stage. We pretrain JoinDet for 5 and 50 epochs on ImageNet-1K and MS COCO, respectively, where the pretraining schedules are set the same as DETReg. We update object priors every 1 and 10 epochs for ImagNet-1K and MS COCO, respectively. As our method relies on the self-attention map in the transformer encoder, most generated object priors focus on big objects, so we use large-scale jittering mentioned in [17] to alleviate the imbalance problem. More implementation details and experimental results are provided in supplementary materials. All experiments are implemented on 16 NVIDIA V100 GPUs.

### 3.2 Comparison with State-of-the-art methods

**Low-Data regimes object detection.** To evaluate the effectiveness of JoinDet, we conduct experiments in a low-data regimes object detection setting on COCO, where only 1% or 10% amounts of COCO labeled data are used during finetuning. The experimental results are presented in Tab. 1. Firstly, our method outperforms existing backbone pretraining methods, with a significant performance gain of **4.68** AP, and **4.53** AP over the supervised pretraining baseline on 1% and 10% COCO data, respectively. Second, when compared with DETReg, which also fully pretrains the model, our method remains 1.31 AP, and 1.75 AP performance gain on 1% and 10% COCO data, respectively, showing the effectiveness of progressively refined object priors.

**Full-data finetuning.** To further evaluate JoinDet, we finetune it on two datasets: MS COCO and PASCAL VOC. On MS COCO, following previous works [57, 3], we finetune JoinDet using a similar training schedule on MS COCO $train2017$ and evaluate on $val2017$. On PASCAL VOC, we finetune on $trainval07 + 12$ and evaluate on $test07$. Comparison with state-of-the art methods is presented in Tab. 2 and Tab. 3. Tab. 2 shows that JoinDet achieves higher performance on VOC. Concretely, JoinDet outperforms DETReg with **+0.2** AP and **+1.0** AP when pretraining on ImageNet-1K and COCO, respectively. When using COCO as the pretraining dataset, JoinDet shows considerable performance gain. We suggest that, as JoinDet is based on mining object priors on the pretraining dataset, more diverse objects contained in COCO provide more diverse supervision on JoinDet, which further boosts its performance when transferred to downstream datasets. Second, JoinDet achieves

Table 2: Object detection results when finetuned on PASCAL VOC. All models are based on Deformable DETR [57].

| Pretraining | Pretrain dataset | Epochs | AP | AP50 | AP75 |
|---|---|---|---|---|---|
| Supervised | ImageNet-1K | 100 | 59.5 | 82.6 | 65.6 |
| SwAV [5] | ImageNet-1K | 100 | 61.0 | 83.0 | 68.1 |
| DETReg [3] | ImageNet-1K | 100 | 63.5 | 83.3 | 70.3 |
| JoinDet | ImageNet-1K | 100 | **63.7** | **83.8** | **70.7** |
| DETReg [3] | COCO | 100 | 63.4 | 83.7 | 70.8 |
| JoinDet | COCO | 100 | **64.4** | **84.0** | **71.3** |

Table 3: Object detection results when finetuned on MS COCO. All models are based on Deformable DETR [57].

| Pretraining | Epochs | AP | AP50 | AP75 |
|---|---|---|---|---|
| Supervised | 50 | 44.5 | 63.6 | 48.7 |
| SwAV [3] | 50 | 45.2 | 64.0 | 49.5 |
| UP-DETR [12] | 50 | 44.7 | 63.7 | 48.6 |
| DETReg [3] | 50 | 45.5 | 64.1 | 49.9 |
| JoinDet | 50 | 45.6 | 64.3 | 49.8 |

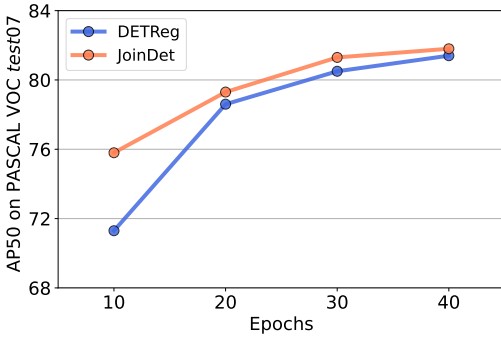

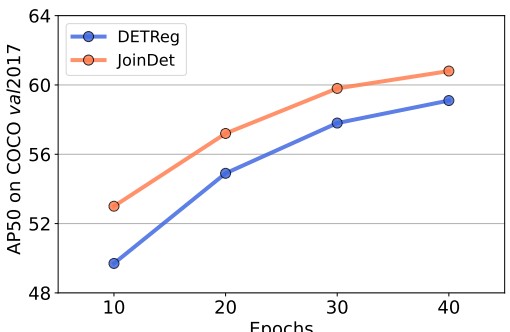

Figure 3: AP50 (COCO style) learning curves with DETReg and JoinDet on VOC when finetuning with small epochs.

Figure 4: AP50 (COCO style) learning curves with DETReg and JoinDet on COCO when finetuning with small epochs.

**+0.4** AP performance gain on SwAV and comparable performance with DETReg on full-data COCO finetuning. Second, as shown in Fig. 3 and Fig. 4, JoinDet is finetuning-efficient, which shows considerable improvement when finetuning with small epochs.

**Class agnostic object proposal evaluation.** To demonstrate that JoinDet learns better to detect objects without labels, following DETReg, we examine the class agnostic performance of JoinDet on Tab. 4. First, JoinDet outperforms other pretraining approaches, indicating that using progressive region priors improves the localization ability. Second, pretraining on COCO shows a larger performance gain of **+1.7** AP, which shows our method benefits more on scene images with various foreground objects.

## 4 Ablation

**Default Settings**. We implement ablations on single-scale deformable DETR. JoinDet is unsupervised pretrained on COCO for 50 epochs with a learning rate drop at epoch 40. The supervision mined from the model outputs is updated every 10 epochs. The results in this section are evaluated by full-data finetuning on PASCAL VOC with 25 epochs. We train 3 different models with different random seeds and report the mean result of AP.

**Select effective object priors.** To assess the effectiveness of joint usage of region priors and instance priors, we compare the performance of JoinDet variants using different object priors: (a) box predictions $\mathcal{P}$, (b) initial region priors $\mathcal{P}_0^r$, (c) region priors $\mathcal{P}^r$, (d) instance priors $\mathcal{P}^i$, (e) region priors $\mathcal{P}^r$ + instance priors $\mathcal{P}^i$. As shown in Tab. 5, using both region and instance priors (Tab. 5(e)) outperforms DETReg by **+3.0** AP, **+1.5** AP with 5 epochs and 25 epochs finetuning, respectively, which verifies the effectiveness of joint usage of two priors generated by the model. Second, when region priors or instance priors are removed, we observe a **-0.6**, **-1.9** AP performance drop with 25 epochs finetuning, respectively, which suggests that region and instance priors can mutually benefit each other. Third, when no region priors are used to initialize the pretraining, directly using the box predictions of the model can only generate unreliable object priors for supervision, leading to a significant performance drop of **-13.4** AP with 25 epochs finetuning.

Table 4: Class agnostic object proposal evaluation on MS COCO $val2017$. Models are pretrained on ImagNet-100 and COCO for 50 epochs. The top 100 proposals are evaluated. Compared with other methods, the higher Average Recall of JoinDet shows that it detects more objects without supervision from the ground truth.

| Method | Pretrain | AP | AP50 | AP75 | AR@1 | AR@10 | AR@100 |
|---|---|---|---|---|---|---|---|
| Selective Search | - | 0.2 | 0.5 | 0.1 | 0.2 | 1.5 | 10.9 |
| UP-DETR [12] | ImageNet-100 | 0.0 | 0.0 | 0.0 | 0.0 | 0.0 | 0.4 |
| DETReg [3] | ImageNet-100 | 1.0 | 3.1 | 0.6 | 0.6 | 3.6 | 12.7 |
| JoinDet | ImageNet-100 | **2.5** | **5.3** | **1.8** | **2.3** | **6.7** | **13.9** |
| DETReg [3] | COCO | 1.3 | 3.0 | 1.0 | 0.6 | 3.5 | 11.7 |
| JoinDet | COCO | **3.0** | **7.4** | **2.3** | **2.3** | **8.8** | **17.4** |

Table 5: Pretrain JoinDet with different object priors. All experiments are pretrained using single-scale Deformable DETR on COCO for 50 epochs and finetuned on VOC.

| Method | Object priors $\mathcal{R}$ from | $\mathcal{P}^r$ from | Update | 10 epochs | 25 epochs |
|---|---|---|---|---|---|
| DETReg | Selective search | - | | 46.0 | 53.9 |
| (a) | $\mathcal{P}$ | - | ✓ | 38.4 | 42.1 |
| (b) | $\mathcal{P}_0^r$ | Backbone | | 46.5 | 53.6 |
| (c) | $\mathcal{P}^r$ | Encoder | ✓ | 46.7 | 53.5 |
| (d) | $\mathcal{P}^i$ | Encoder | ✓ | 48.1 | 54.8 |
| (e) | $\mathcal{P}^r \oplus \mathcal{P}^i$ | Encoder | ✓ | **49.0** | **55.4** |

**Smooth the refinement of object priors.** To demonstrate the effectiveness of the proposed Box Smooth Module, we conduct two other updating methods, including directly changing and updating by NMS (see Fig. 5). Directly changing, which updates the supervision without considering previous generated object priors, leads to a fast change on the supervision signal. When updating by NMS, object priors are only updated when they have a large IoU with current object priors, making a relatively slow updating speed. As summarized in Tab. 6, noticeable performance drops are observed without the Box Smooth Module, especially when the updating method is altered by directly changing, which verifies the effectiveness of the Box Smooth Module. The large performance gaps are due to the facts the: (1) Directly changing neglects useful object priors in previous supervision, which makes object-related object priors appear and disappear during pretraining, leading to an unstable training process; (2) Although updating by NMS avoids the disappearance of reliable object priors in previous supervision, it fails to refine the box coordinates in a slow and smooth way. The quick shifting of the supervision signal leads to undesirable learning instabilities.

**Augmentations during pretraining.** As our method mines object priors on the self-attention map in the transformer encoder, most generated priors focus on big objects. To assess the importance of small object priors during pretraining, we ablate on different augmentations during pretraining. We use the initial supervision $\mathcal{R}_0$, which bounds the highlighted regions in the eigen attention map generated by the backbone as fixed supervision. All experiments are pretrained over 50 epochs on COCO with Deformable DETR and evaluated on PASCAL VOC with 100 epochs finetuning. It can be observed in Tab. 7 that both scale jittering and copy & paste improve the pretraining with a large margin, indicating the importance of adding small object priors when mining supervision from the self-attention map.

## 5  Visualization

To qualitatively evaluate the generated object priors in our method, we visualize the evolution of the eigen attention map and generated object priors during pretraining, and compare them with that in a supervised trained Deformable DETR in Fig. 6. First, generated object priors are progressively refined to objects in images during pretraining. Second, supervised by updating object priors, foregrounds in the eigen attention map can be further highlighted (see Fig. 6(a)). Third, with the help of progressive instance priors, the generated object priors can still become object-related and similar to ground truth

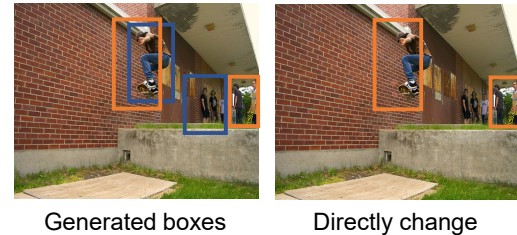 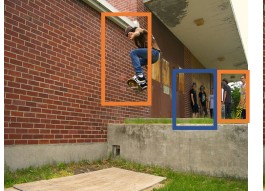 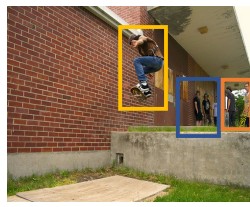

| Generated boxes | Directly change | Update by NMS | Box Smooth Module |
|---|---|---|---|

☐ Previous reliable object priors $\mathcal{R}_{n-1}$ ☐ Current reliable object priors $\mathcal{R}_n$

Figure 5: Qualitative samples for different object priors updating methods. Directly changing updates the supervision without considering previous object priors. Updating by NMS reserves previous object priors which have small IoUs with current object priors. Box Smooth Module suppresses inaccurate previous object priors and slows down the shifting speed of object priors.

Table 6: VOC finetuning results of different object priors updating methods during pretraining. All methods are pretrained on COCO without labels.

| Method | Change | 10 epochs | 25 epochs |
|---|---|---|---|
| Directly change | Fast | 48.0 | 53.0 |
| Update by NMS | Mid | 48.3 | 54.5 |
| Box Smooth Module | Slow | **49.0** | **55.4** |

Table 7: VOC finetuning results of different augmentations during pretraining. All methods are pretrained on COCO without labels.

| Method | AP |
|---|---|
| Using the initial supervision $\mathcal{R}_0$ | 62.3 |
| +scale jittering [17] | 63.4 |
| +copy & paste [17] with small boxes | 63.5 |

even when it is hard for the eigen attention map to be further refined, *e.g.* when foreground objects fill images (Fig. 6(b)) or objects appear in a simple background (Fig. 6(c)). In the former condition, our method highlights all objects while the supervised method focuses on labeled people. In the latter condition, foregrounds and backgrounds have been well divided at the beginning epochs.

## 6 Related Works

**Unsupervised pretraining for backbone.** Recent unsupervised pretraining methods, which rely on pretext tasks to learn visual representations [9, 21, 18, 7, 10, 8, 44, 49, 33, 54, 55, 43], have shown considerable performance on transfer learning tasks, outperforming their supervised counterparts. However, compared with considerable performance gains on classification-related tasks, the improvement on dense-prediction tasks [30, 16] is limited. To this end, a growing number of works explore pretext tasks for object detection and instance segmentation. DenseCL [46] and PixPro [53] contrast pixel feature on the same physical location under different views to learn pixel-level representations. DetCo [51] exploits supervision on features from different stages of the backbone and from global and local patches to learn consistent representations on image-level and patch-level. [2] proposes point-level region contrast, which enables the model to learn at the point-level to help localization, and at the region-level to help holistic object recognition. Despite the good performance, all these works only focus on pretraining the backbone of object detectors, neglecting the detector heads. When these methods are transferred to object detection, the detector heads are initialized from scratch and do not benefit from pretraining, which limits their performance on object detection. In contrast, our JoinDet, which utilizes object priors generated by the model itself as supervision, pretrains the entire model to promote detector learning.

**Unsupervised pretraining for object detector.** Pretraining the backbone with a pretext task for dense-prediction tasks leaves untrained detection heads which are also a core component when transferring to object detection [24]. Few works attempt to remedy this problem by pretraining the entire detector with various unsupervised pretext tasks. SoCo [48] utilizes selective search to generate object priors and perform contrastive learning on object-level features from the detector head. UP-DETR [12] and DETReg [3] pretrain the detection heads of DETR [4] by forcing them to predict object priors generated by randomly cropping and selective search, respectively. However, randomly cropping hardly provides any effective object prior, and selective search is a heuristic method that is time-consuming, independent from the pretraining process. In contrast to these methods, our

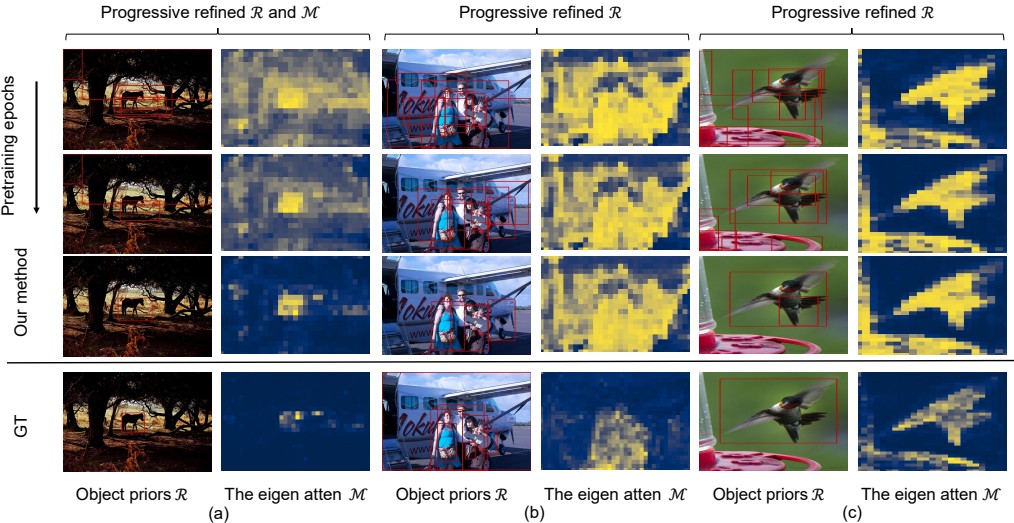

Figure 6: Visualization results of generated object priors and the eigen attention map of JoinDet on COCO. In (a), we show that generated object priors and the eigen attention maps can be progressively and mutually refined. In (b) and (c), generated object priors are still refined when it is hard for the eigen attention maps to be further refined during the pretraining process.

proposed JoinDet jointly generates object priors and learns detection, which can gradually update the object priors with learned and improved ones for better supervision during pretraining.

**Attention in unsupervised pretraining as supervision.** NNCLR [15] and DINO [6] show that the attention maps of the visual transformer can generate semantic segmentation masks even though the model is pretrained without labels. This suggests that self-learned attention can provide effective supervision for dense-prediction tasks. STEGO [19] utilizes off-the-shelf pretrained DINO to extract cross-image feature correspondence (cross-image attention) as supervision to distill segmentation features and train an unsupervised segmentation model. Different from STEGO, our JoinDet exploits the self-attention maps in the transformer encoder to generate multiple object priors as supervision during training. We also show that the self-attention maps can be jointly refined during training to generate progressive object priors for better supervision.

# 7 Conclusion

In this paper, we mine the information from the knowledge learned by the object detection model itself to pretrain the model without labels. We propose an unsupervised object detection pretraining method which can jointly generate reliable object priors and learn to detect objects. The core is at generating region priors from the transformer encoder and instance priors from the transformer decoder to achieve reliable object priors for supervision. With careful design, the model can gradually learn better object localization capability, producing better object priors to optimize itself. Our method shows considerable improvements in low-data regimes object detection and fast convergence on full-data finetuning. However, there still exists a large gap from supervised training on class agnostic object proposal evaluation, which calls for further studies.

# 8 Acknowledgment

This work is supported by the Key R&D Project of Zhejiang Province (No.2022C01056) and the National Natural Science Foundation of China (No.62127803).

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

# A    Appendix

Optionally include extra information (complete proofs, additional experiments and plots) in the appendix. This section will often be part of the supplemental material.

