# Unsupervised Object Detection Pretraining with Joint Object Priors Generation and Detector Learning

**Yizhou Wang[1,3]*    Meilin Chen[1,3]*    Shixiang Tang[2] †    Feng Zhu[3]    Haiyang Yang[5]**
**Lei Bai[4]    Rui Zhao[3,6]    Yunfeng Yan[1]    Donglian Qi[1]    Wanli Ouyang[4,2]**
[1]Zhejiang University    [2]The University of Sydney    [3]SenseTime Research
[4]Shanghai AI Laboratory    [5]Nanjing University
[6]Qing Yuan Research Institute, Shanghai Jiao Tong University, Shanghai, China
{yizhouwang, merlinis, yvonnech, qidl}@zju.edu.cn    stan3906@uni.sydney.edu.au
{zhufeng, zhaorui}@sensetime.com    baisanshi@gmail.com
hyyang@smail.nju.edu.cn    wanli.ouyang@sydney.edu.au

## 1  More experimental results

One key factor that contributes to the success of JoinDet is using progressively refined object priors as supervision. We have already shown that the selection of effective object priors have a huge impact on finetuning performance in Sec.4 of the main text. Here, we provide more experimental results to explore the influence of hyperparameters in JoinDet and discuss the possible direction for future works. We implement experiments on single-scale deformable DETR [11]. Unless otherwise specified, we set the momentum coefficient in the Box Smooth Module as 0.45, and the clustering IoU threshold in the Box Smooth Module as 0.48. The supervision generated from object priors is updated every 10 epochs by default. JoinDet is pretrained on COCO for 50 epochs and finetune on VOC for 25 epochs. We train 3 different models with different random seeds and report the mean result of AP (COCO format) on VOC.

### 1.1  Momentum coefficient in the Box Smooth Module

The momentum coefficient $m^s$ in Box Smooth Module controls the shifting speed of supervision, which considers both precedent object priors and current object priors. We ablate the most suitable momentum coefficient for JoinDet in Tab. 1. Firstly, small momentum coefficients, which are smaller than 0.45, represent a relative fast-shifting speed of supervision, showing significant performance drops. Concretely, when $m^s = 0$, the supervision is directly replaced with current object priors, neglecting useful precedent object priors and leading to **-2.4** AP drop. Second, when the shifting speed is too slow ($m^s = 0.70$), behindhand object priors are insufficient to guide the current model, which is also harmful (55.4 AP→53.7 AP) for JoinDet.

### 1.2  Clustering IoU threshold in the Box Smooth Module

When precedent object priors and current object priors have large IoUs, which are bigger than the threshold, corresponding priors (boxes) will be clustered in the same cluster. The box coordinates and scores of all boxes in a specific cluster will be used to generate a new box for supervision. Experimental results of using different clustering IoU thresholds are summarized in Tab. 2. First, we find 0.48 as an optimal hyperparameter, suggesting that duplicate object priors with larger thresholds and scarce object priors with smaller thresholds are both harmful for pretraining. Second, the performance variation with different cluster IoU thresholds are relatively slight (at most -1.3 AP), which indicates that our proposed method is robust to the clustering IoU thresholds.

---

*Equal contribution. The work was done during an internship at SenseTime.
†Corresponding author.

Table 1: Pretrain JoinDet with different momentum coefficients. When momentum coefficient $m^s = 0$, the supervision will be directly changed to current object priors. AP on VOC is reported.

| Method | ms | 10 epochs | 25 epochs |
|--------|------|-----------|-----------|
| DETReg | - | 46.0 | 53.9 |
| JoinDet | 0.70 | 47.1 | 53.7 |
| | 0.45 | **49.0** | **55.4** |
| | 0.20 | 48.3 | 54.3 |
| | 0.05 | 47.7 | 54.3 |
| | 0 | 48.0 | 53.0 |

Table 2: Pretrain JoinDet with different clustering IoU thresholds. AP on VOC is reported.

| Method | IoU threshold | 10 epochs | 25 epochs |
|--------|---------------|-----------|-----------|
| DETReg | - | 46.0 | 53.9 |
| JoinDet | 0.35 | 46.2 | 54.1 |
| | 0.40 | 47.1 | 53.9 |
| | 0.48 | **49.0** | **55.4** |
| | 0.55 | 48.1 | 54.8 |
| | 0.60 | 48.4 | 54.9 |
| | 0.65 | 47.9 | 54.8 |

## 1.3 Update frequency

As generated object priors are progressively refined during pretraining, we update object priors every 10 epochs as the supervision. As shown in Tab. 3, when the momentum coefficient is fixed (0.45), updating the supervision too frequently (every 1 epoch) leads to a significant performance drop, which indicates that stable supervision is very important to unsupervised pretraining for object detectors. We argue that the performance drop brought by frequent updating can be remedied with a proper momentum coefficient as discussed in Sec.2 of the main text, which we remain for future work.

Table 3: Pretrain JoinDet with different update frequencies. AP on VOC is reported.

| Method | Update frequency | 10 epochs | 25 epochs |
|--------|------------------|-----------|-----------|
| DETReg | - | 46.0 | 53.9 |
| JoinDet | 1 epoch | 40.7 | 51.3 |
| | 5 epochs | 46.6 | 54.0 |
| | 10 epochs | **49.0** | **55.4** |
| | 20 epochs | 46.8 | 54.6 |

## 1.4 Comparison with supervised training

Lots of previous papers [1,2,3] in this field use supervised pretraining on ImageNet as a baseline pretraining method and ignore the difference of pertaining epochs between supervised and unsupervised methods. In this paper, to further explore the effectiveness of JoinDet, we add more finetuning epochs to make the supervised method have similar training epochs (counting both pretraining epochs and finetuning epochs) as JoinDet.

As shown in Tab. 4, we extend the training epochs to 200 epochs for the supervised pretraining on PASCAL VOC and achieve 59.3 AP, which is still lower (-5.1AP) than our JoinDet. On the relatively small dataset, PASCAL VOC, the supervised pretraining shows to be over-fitting with 200 epochs. We suggest that the performance gap between supervised pretraining and JoinDet verifies the effectiveness of detection pre-training. In the special case where pretraining data and fine-tuning are exactly the same in COCO, extending the training epochs for supervised pretraining results in a comparable performance as JoinDet. However, we suggest that fine-tuning using less data or

fewer epochs is a more common and valuable setting because it is closer to real needs and collecting larger-scale unlabeled pretraining data through the Internet is easy and feasible.

Table 4: Compare with the supervised method using more full-data finetuning epochs.

| Finetune dataset | Methods | Pretrain on COCO without labels | Full-data Finetune | AP |
|---|---|---|---|---|
| VOC | Supervised | 0 | 100 | 59.5 |
| | Supervised | 0 | 200 | 59.3 |
| | JoinDet | 50 | 100 | 64.4 |
| COCO | Supervised | 0 | 50 | 44.5 |
| | Supervised | 0 | 100 | 45.6 |
| | JoinDet | 50 | 50 | 45.6 |

## 1.5 Combine Selective Search regions

There is an intuition that Selective Search can provide some meaningful regions using low-level cues, however, simply introducing Selective Search regions leads to performance drops on downstream tasks. As shown in Tab. 5 following the ablation setting in Sec. 4 in the main text, we add 30 selective search boxes with originally generated object priors as supervision during pretraining and find that additional selective search proposals lead to -2.5AP and -0.5AP performance drops with 10 epochs and 25 epochs finetuning on VOC, respectively. We suggest that there are two reasons. (1) Low-level cues lack semantic information and will introduce lots of non-object-related supervision, which is harmful to detector pretraining when more accurate self-supervised information is presented (see in Fig. 1). (2) JoinDet is already able to generate useful supervision for small objects and objects that have clear boundaries to the background. We use large-scale jittering mentioned in [5] to provide supervision for small objects.

Table 5: Incorporate Selective Search regions with object priors generated by JoinDet.

| Method | 10 epochs VOC | 25 epochs VOC |
|---|---|---|
| JoinDet | 49 | 55.4 |
| JoinDet + Selective Search | 46.5 | 54.8 |

## 2 Additional visualization

Fig. 1 visualizes more progressively refined object priors by JoinDet and fixed object priors by selective search. For select search, we only visualize the top 15 object priors. JoinDet generates object priors with fewer background regions than selective search.

## 3 The eigen attention map computation method $\mathcal{K}$

According to [10], the eigen attention map in the vision transformer can highlight salient foregrounds by partitioning all features $\mathbf{f}_i \in \mathbb{R}^c$ in output patch features $\mathcal{F} \in \mathbb{R}^{h \times w \times c}$ into the background set $\mathcal{F}^b$ and the foreground set $\mathcal{F}^f$, where $i \in [1, hw]$, and $h$, $w$, $c$ denote the height, width, and dimension of output patch features $\mathcal{F}$, respectively. Following [10, 9], we fix the feature partition task by solving a group partition problem on a self-similarity graph $\mathcal{S} = (\mathcal{V}, \mathcal{U})$, where the nodes $\mathcal{V}$ represent all features on $\mathcal{F}$ and the edges $\mathcal{U}$ are based on the cosine similarity between corresponding features, which can be computed by

$$\mathcal{U}_{i,j} = \begin{cases} 1, & \text{if } \cos(\mathbf{f}_i, \mathbf{f}_j) \geq \tau \\ \epsilon, & \text{otherwise} \end{cases},$$

$$\cos(\mathbf{f}_i, \mathbf{f}_j) = \frac{\langle \mathbf{f}_i, \mathbf{f}_j \rangle}{\|\mathbf{f}_i\|_2 \cdot \|\mathbf{f}_j\|_2},$$

(1)

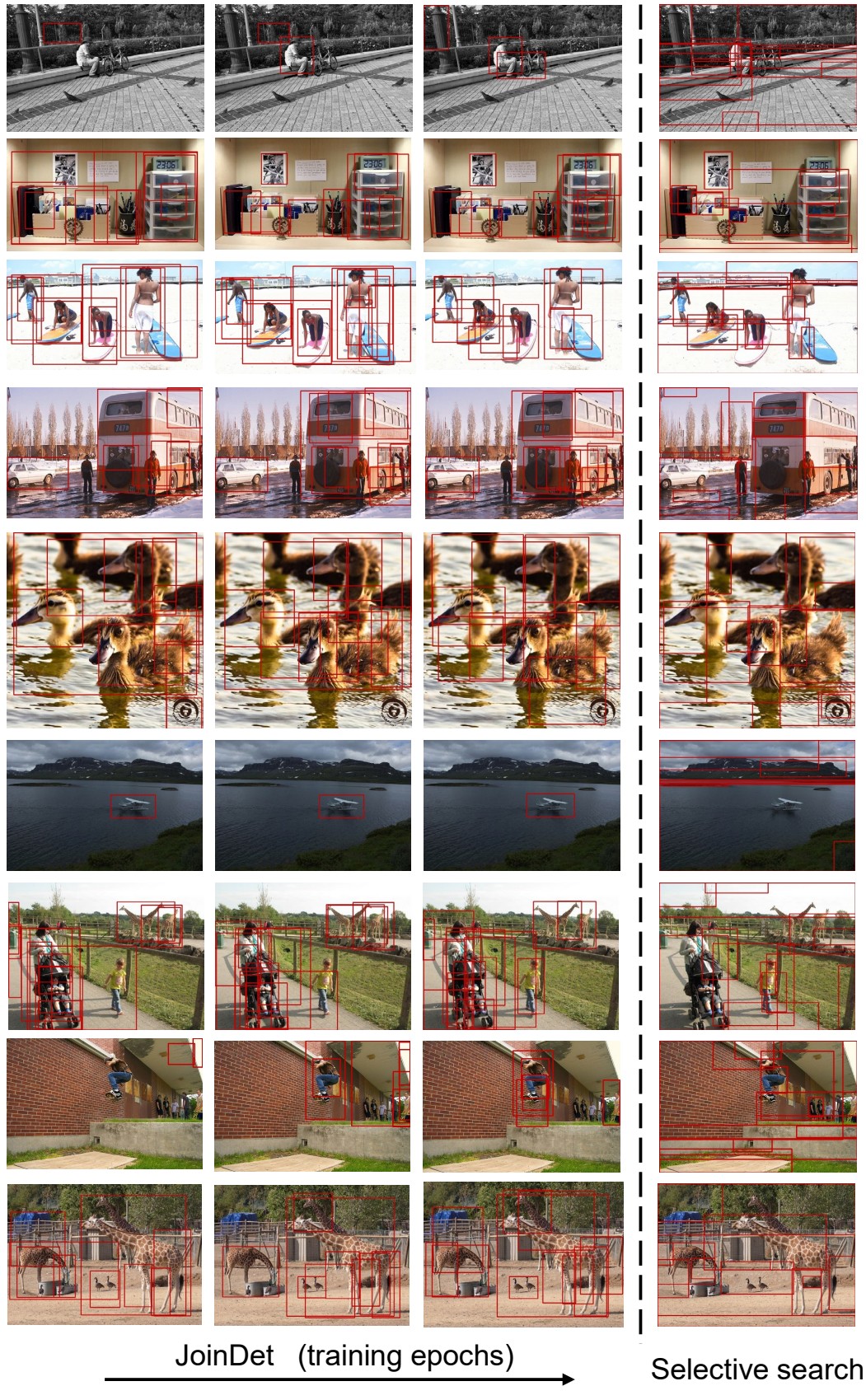

JoinDet   (training epochs)

Selective search

Figure 1: Evolution of object priors generated by JoinDet and object priors generated by Selective Search. We show that progressively refined object priors in JoinDet contain fewer background regions.

where $\mathcal{U}_{i,j}$ denotes the edge between feature $\mathbf{f}_i$ and feature $\mathbf{f}_j$, $\cos$ denotes the cosine similarity, $\tau$ is a hyper-parameter and $\epsilon$ equals a small positive value to ensure that the graph is fully-connected. To partition the graph $\mathcal{S}$ into two disjoint sets $\mathcal{F}^f$ and $\mathcal{F}^b$, we simply remove edges connecting the two parts. The optimal bi-partitioning of the graph $\mathcal{S}$ can be solved by minimizing the Ncut energy $\mathbb{E}$ [9, 10]:

$$\min_{\mathcal{F}^f, \mathcal{F}^b} \mathbb{E}(\mathcal{F}^f, \mathcal{F}^b) = \min_{\mathcal{F}^f, \mathcal{F}^b} \left[ \frac{C(\mathcal{F}^f, \mathcal{F}^b)}{C(\mathcal{F}^f, \mathcal{V})} + \frac{C(\mathcal{F}^f, \mathcal{F}^b)}{C(\mathcal{F}^b, \mathcal{V})} \right], \tag{2}$$

where $C(\mathcal{F}^b, \mathcal{F}^f) = \sum_{\mathbf{u} \in \mathcal{F}^b, \mathbf{t} \in \mathcal{F}^f} \mathcal{U}_{\mathbf{u}, \mathbf{t}}$ measures the degree of similarity between two sets. By reducing Eq. 2, maximizing the similarity within the sets and minimizing the dissimilarity between two sets can be satisfied simultaneously [9].

Let $\mathbf{1}$ be an vector of all ones, and $\mathbf{x}$ be an dimensional indicator vector, $\mathbf{x}_i = 1$ if node $i$ is is in $\mathcal{F}^f$ and -1, otherwise. Indicating in [9], the optimization problem in Eq. 2, which is NP-complete, can be equivalently substituted by

$$\min_{\mathbf{x}} \mathbb{E}(\mathbf{x}) = \min_{\mathbf{y}} \frac{\mathbf{y}^T (\mathbf{D} - \mathcal{U}) \mathbf{y}}{\mathbf{y}^T \mathbf{D} \mathbf{y}}, \tag{3}$$

where $\mathbf{D}$ is a diagonal matrix with total connection from node $i$ to all other nodes $\mathbf{d}(i) = \sum_j \mathcal{U}_{i,j}$ on its diagonal, $\mathbf{y} \in \{1, -b\}$ and $b$ satisfies $\mathbf{y}^T \mathbf{D} \mathbf{1} = 0$.

Eq. 3 is the Rayleigh quotient [6]. If $\mathbf{y}$ is relaxed to take on real values, Eq. 3 can be minimized by solving

$$(\mathbf{D} - \mathcal{U}) \mathbf{y} = \lambda \mathbf{D} \mathbf{y}. \tag{4}$$

Let $\mathbf{z} = \mathbf{D}^{-\frac{1}{2}} \mathbf{y}$, we can rewrite Eq. 4 as

$$\mathbf{D}^{-\frac{1}{2}} (\mathbf{D} - \mathcal{U}) \mathbf{D}^{-\frac{1}{2}} \mathbf{z} = \lambda \mathbf{z}. \tag{5}$$

And the energy in 3 can be rewritten as

$$\min_{\mathbf{z}} \frac{\mathbf{z}^T \mathbf{D}^{-\frac{1}{2}} (\mathbf{D} - \mathcal{U}) \mathbf{D}^{-\frac{1}{2}} \mathbf{z}}{\mathbf{z}^T \mathbf{z}}. \tag{6}$$

It can be easily proofed that $\mathbf{z}_0 = \mathbf{D}^{-\frac{1}{2}} \mathbf{1}$ is an eigenvector of Eq. 5 with eigenvalue of 0, which satisfied the constraint $\mathbf{y}^T \mathbf{D} \mathbf{1} = 0$. As $(\mathbf{D} - \mathcal{U})$, called the Laplacian matrix, is positive semidefinite, $\mathbf{D}^{-\frac{1}{2}} (\mathbf{D} - \mathcal{U}) \mathbf{D}^{-\frac{1}{2}}$ is symmetric positive semidefinite [8]. Therefore $\mathbf{z}_0$ is the smallest eigenvector of Eq. 5, and $\mathbf{z}_1$, the second smallest eigenvector of Eq. 5, is perpendicular to $\mathbf{z}_0$ [9]. According to the Rayleigh quotient [6], $\mathbf{z}_1$, the second smallest eigenvector of Eq. 5, is the real valued solution to minimize the energy in Eq. 6,

$$\mathbf{z}_1 = \arg\min_{\mathbf{z}^T \mathbf{z}_0} \frac{\mathbf{z}^T \mathbf{D}^{-\frac{1}{2}} (\mathbf{D} - \mathcal{U}) \mathbf{D}^{-\frac{1}{2}} \mathbf{z}}{\mathbf{z}^T \mathbf{z}}. \tag{7}$$

Consequently, taking $\mathbf{z} = \mathbf{D}^{-\frac{1}{2}} \mathbf{y}$,

$$\mathbf{y}_1 = \arg\min_{\mathbf{y}^T \mathbf{D} \mathbf{1} = 0} \frac{\mathbf{y}^T (\mathbf{D} - \mathcal{U}) \mathbf{y}}{\mathbf{y}^T \mathbf{D} \mathbf{y}}. \tag{8}$$

Therefore, $\mathbf{y}_1$, the second smallest eigenvector of Eq. 4, is the real valued solution that achieves the optimal partition with Ncut energy $\mathbb{E}$ in Eq. 2.

We then reshape the second smallest eigenvector $\mathbf{y}_1$ to the eigen attention map $\mathcal{M} \in \mathbb{R}^{h \times w}$, which has the same height and width with output patch features $\mathcal{F}$.

## 4 Training Details

### 4.1 Pretraining

Following DETReg [1], we initialize the ResNet50 backbone of JoinDet with SwAV [2], which was pretrained on ImageNet1K [3] for 800 epochs, and fix the backbone during pretraining. Furthermore, a same SwAV encoder is used to extract features of object priors, which are cropped and resized

to $128 \times 128$. JoinDet follows the default hyperparameter setting and training strategy used in Deformable DETR [11], except that the object embedding loss with loss weight 1. On COCO [7], models are trained for 50 epochs and the learning rate is decayed by a factor of 0.1 at epoch 40. On ImageNet [3], following DETReg [1], we train models for 5 epochs. Following Deformable DETR [11], we train our models using the Adam optimizer with a base learning rate of $2 \times 10^{-4}$, $\beta_1 = 0.9$, $\beta_2 = 0.999$, and set the weight decay as $10^{-4}$. We use large scale jittering mentioned in [5] as an additional augmentation to alleviate the scale imbalance problem in generated object priors.

### 4.2 Evaluation

We finetune JoinDet on COCO [7], VOC [4] to evaluate our method. When finetuning, the original classification branch $f_{cls}$ and the object embedding branch is dropped. We initiate a new classification branch using a single fully-connect layer with output dimension $c$, where $c$ denotes the total categories in the downstream detection datasets.

**Full-data finetuning**. For COCO, we finetune models for 50 epochs and the learning rate is decayed by a factor of 0.1 at the 40-th epoch. For VOC, following DETReg [1], models are trained for 100 epochs with the learning rate decayed by a factor of 0.1 at the 70-th epoch.

**Low-Data regimes object detection.** Following DETReg [1], we finetune JoinDet with 1%, 10% COCO training set data with 2000 epochs, 400 epochs, respectively. The base learning rate is set as $2 \times 10^{-4}$ and the learning rate is decayed by a factor of 0.1 at the 1400-th epoch, the 280-th epoch, respectively.

## 5 Broader impact

We present a more effective general unsupervised object detection pretraining method that can jointly generate object priors and learn to detect. Compared with supervised learning, our method eases the burden of expansive and time-consuming manual labels and benefits from rapidly increasing real-world data. Meanwhile, our method can promote the development on smart healthcare because it can be directly used on medical images without labeling by expertise.

However, several potential issues should be taken into consideration when applied in real-world scenarios. First, similar to other learning methods, there still remain concerns about interpretability and robustness. Second, pretrained on manually collected datasets, the method might learn biased features when given with biased datasets. Finally, like other unsupervised pretraining methods, our method relies on extra epochs to pretrain the model, which is not efficient during pretraining, leading to more electricity consumption.