# OpenReview forum: "Unsupervised Object Detection Pretraining with Joint Object Priors Generation and Detector Learning"
_NeurIPS.cc/2022/Conference — NeurIPS 2022 Accept_

### Official Review · Reviewer_fpzy · 2022-07-10

**Rating:** 4
**Confidence:** 4
**Soundness:** 3 good
**Presentation:** 3 good
**Contribution:** 2 fair

**Summary:**

This paper proposes an unsupervised object detection pre-training framework that can generate object priors and learn detector jointly. This work is inspired by DINO, the self-attention maps can generate region prior bounding boxes. Authors utilize Deformable DETR as their main framework. The proposed method achieves good results on low data-regime object detection on COCO and full-data finetuing on PASCAL VOC.

**Questions:**

My biggest concern comes from the experiments and the limited improvements under fine-tuning setting, see weaknesses above.

**Limitations:**

Authors have discussed broader impacts in their supplementary material.

**Strengths And Weaknesses:**

Strengths:
1) The motivation is good. Generating object priors by attention maps is an alternative of the popular selective search algorithm. However, a pre-trained backbone is important and necessary.

Weaknesses:
1) In Section 3, authors should claim which detection framework they used for pre-training in the begging of section 3, which will make the paper clearer.
2) The proposed method highly rely on the pre-trained backbone, authors initialize the ResNet50 backbone of JoinDet with SwAV, did authors explore other unsupervised pre-training methods, e.g. BYOL, SimCLR?
3) Authors should highlight the setting of the pre-training methods, supervised/SwAV only pre-trains backbone, while JoinDet pre-trains whole network.
4) There are experiments conducted in the setting where pre-training is performed on COCO/ImageNet and finetuning is performed on PASCAL VOC (Table2). Why authors did not do experiments in the setting where pre-training is conducted on ImageNet and fine-tuning is performed on COCO. I hope to see this experiment in the rebuttal.
5) Why authors only do experiments under low-data regime on COCO, is it consistent on PASCAL VOC?
6) The improvement over DETReg is somewhat limited.

---

> ### Author Response · Authors · 2022-08-02
> **Response to Reviewer fpzy (part i of ii)**
>
> >**Q1:** In Section 3, authors should claim which detection framework they used for pre-training in the begging of section 3, which will make the paper clearer. Authors should highlight the setting of the pre-training methods, supervised/SwAV only pre-trains backbone, while JoinDet pre-trains whole network.
>
> **A1:** Thanks for the reviewer's writing suggestions. We will revise these parts in our future version to make the paper clearer.
>
> >**Q2:** The proposed method highly rely on the pre-trained backbone, authors initialize the ResNet50 backbone of JoinDet with SwAV, did authors explore other unsupervised pre-training methods, e.g. BYOL, SimCLR?
>
> **A2:** Following the ablation setting in Sec. 4, we change the pretraining from SwAV to BYOL and find that BYOL brings more improvements, which **further demonstrates the compatibility of our method** to backbone pretraining methods and the effectiveness of the detector pretraining paradigm. As shown in the table below, JoinDet using BYOL achieves a **+0.6 AP** performance gain in the ablation study. We only use SwAV as the backbone pretraining method in this paper for making a fair comparison because previous works (UP-DETR[1], DETReg[2]) conduct their experiments using the SwAV pretraining. There may be some other powerful unsupervised pre-training methods for this task, but it is out of the paper's scope and we will leave them for future works.
>
> | Method                  | 10 epochs VOC | 25 epochs VOC |
> |-------------------------|:-------------:|:-------------:|
> | JoinDet(SwAV) - default |      49.0     |      55.3     |
> | JoinDet(BYOL)           |      **49.8**     |      **55.9**     |
>
> >**Q3:** There are experiments conducted in the setting where pre-training is performed on COCO/ImageNet and finetuning is performed on PASCAL VOC (Table2). Why authors did not do experiments in the setting where pre-training is conducted on ImageNet and fine-tuning is performed on COCO. I hope to see this experiment in the rebuttal.
>
> **A3:** Thanks for your review. (1) **Reasons:** our JoinDet mainly focuses on pretraining detectors on scene images that contain multiple objects, which are easier to obtain on the Internet. The region priors can highlight the foreground regions and instance priors can predict multiple object bounding boxes from foreground regions. However, ImageNet is an object-centric dataset, which mostly contains only one object, on which our design is not specifically targeted. As shown in Tab. 2, JoinDet pretrained on COCO performs better (+0.7 AP) than JoinDet pretrained on ImageNet when evaluated on PASCAL VOC because JoinDet can **mine more diverse objects** in the pretraining dataset.
>
> (2) **Experiments:** following your suggestion, we pretrain our model on ImageNet-1K and fine-tune it on COCO. The performance improvement still EXISTS, i.e., JoinDet improves DETReg for **+0.46 AP** and **+0.69 AP** on 1% and 10% COCO fine-tuning settings, respectively.
> | Method  | Pretraining dataset | 1% COCO | 10% COCO |
> |---------|:-------------------:|:-------:|:--------:|
> | DETReg  |     ImageNet-1K     |  14.76  |   29.36  |
> | JoinDet |     ImageNet-1K     |  **15.22**  |   **30.05**  |
>
> >**Q4:** Why authors only do experiments under low-data regime on COCO, is it consistent on PASCAL VOC?
>
> **A4:** (1) We did not do the low-data regime PASCAL VOC experiments because PASCAL VOC is already a **relatively small** downstream detection dataset when compared with COCO. Concretely, PASCAL VOC has only 20K images which are about **1/6** of the images in COCO. And previous works (UP-DETR[1], DETReg[2]) did not do these experiments.
>
> (2) Following your suggestions, we do experiments on PASCAL VOC in low-data regimes. **Consistent performance improvement** can be found in the low-data regime object detection setting on Pascal VOC. As shown in the table below, compared with DETReg, JoinDet achieves **+2.29 AP**, and **+1.60 AP** performance gains on 1%, and 10% VOC fine-tuning settings.
> | Method  |   Pretrain  |    1% VOC    |    10% VOC   |
> |---------|:-----------:|:------------:|:------------:|
> | SwAV    |   Backbone  |     14.02    |     33.80    |
> | DETReg  | Whole model |     21.12    |     44.37    |
> | JoinDet | Whole model | **23.41**(+2.29) | **45.97**(+1.60) |

---

> > ### Author Response · Authors · 2022-08-02
> > **Response to Reviewer fpzy (part ii of ii)**
> >
> > >**Q5:** The improvement over DETReg is somewhat limited.
> >
> > **A5:** JoinDet shows considerable performance gain on **three evaluation benchmarks** in unsupervised learning, e.g., low-data regimes object detection on COCO, few-epochs full-data finetuning on COCO, and full-data fine-tuning on PASCAL VOC. Specifically, when 10% COCO data are used for fine-tuning, JoinDet shows a **+1.75 AP** performance gain on DETReg (Tab. 1). When fine-tuning with fewer (10) epochs on COCO, DETReg achieves a **+3.3 AP50** performance gain when compared with DETReg (shown in Fig. 4). On full-data PASCAL VOC fine-tuning, JoinDet improves DETReg by **+1.0 AP** (Tab. 2).
> >
> > Only the improvement on the special full-data fine-tuning on COCO is +0.1% (Tab. 3).  In this **special** setting, the pretraining data and fine-tuning data are **exactly** the same, and the pretrained models are finetuned by **sufficiently long epochs** (i.e., deformable DETR almost converges with 50 epochs training [3]).  We consider this setting unsuitable to evaluate the performance of pretrainings because sufficiently long epochs eliminate the performance difference among different pretrainings. More importantly, when we extend the fine-tuning epochs for supervised pretraining to 100 epochs, we get a detection performance of 45.6 AP. Considering our JoinDet reaches **45.6 AP** using **only** 50 epochs of fine-tuning, we suggest that **JoinDet has reached the upper bound of deformable DETR on COCO** when we fine-tune the pretrained model with 50 epochs. We provide these results just for making the evaluation more comprehensive following existing works.
> >
> > We would call attention to one more common evaluation setting where **fine-tuning data is less or epochs are fewer** as the "pretraining - finetuning" paradigm always uses a large dataset to pretrain and small epochs to finetune. Because it is closer to real needs and collecting larger-scale unlabeled pretraining data through the Internet is easy and feasible.
> > | Finetune dataset |   Methods  | Pretrain epochs on COCO without labels | Full-data Finetune epochs |  AP  |
> > |:----------------:|:----------:|:-------------------------------:|:------------------:|:----:|
> > |       COCO       | Supervised |                0                |         50         | 44.5 |
> > |       COCO       | Supervised |                0                |         100        | 45.6 |
> > |       COCO       |   JoinDet  |                50               |         50         | 45.6 |
> >
> > **Reference:**
> >
> > [1] Dai, Zhigang, et al. "Up-detr: Unsupervised pre-training for object detection with transformers." Proceedings of the IEEE/CVF conference on computer vision and pattern recognition. 2021.
> >
> > [2] Bar, Amir, et al. "Detreg: Unsupervised pretraining with region priors for object detection." Proceedings of the IEEE/CVF Conference on Computer Vision and Pattern Recognition. 2022.
> >
> > [3] Zhu, Xizhou, et al. "Deformable DETR: Deformable Transformers for End-to-End Object Detection." International Conference on Learning Representations. 2020.

---

### Official Review · Reviewer_34gJ · 2022-07-10

**Rating:** 5
**Confidence:** 4
**Soundness:** 3 good
**Presentation:** 3 good
**Contribution:** 2 fair

**Summary:**

This submission proposes a pre-training approach for object detection transformer, it generates pseudo bounding box regression targets by leveraging the attention cues from the pre-trained backbone network. Specifically, both object proposals generated merely in encoder attention map and object instance prediction from decoder head are used for pseudo boxes. The method is evaluated on low-data and full data fine-tuning, as well as object proposal generation.

**Questions:**

1. I actually have a question regarding the effectiveness of detection pre-training. For the **Full-data finetuning** experiment, the supervised training is compared as a baseline, however, would it be fair to add the pre-training time to the baseline? I guess when you extend the baseline training time, it would easily catch up with the detection pre-training methods, which would pose a concern on the effectiveness of detection pre-training for full-data finetuning.
2. Although the work does not employ low-level unsupervised proposal methods like Selective Search, would it be more advantageous to incorporate these proposals which are generated from low-level cues? The intuition is that they may provide better or complementary coverage to some objects such as small ones, or others that have clear boundaries to the background.
3. I'm interested in how the pre-trained object feature embedding loss affects the method, have you tried removing the feature embedding loss? Do you have any perspectives, e.g., it is a regularization that avoids the feature representation deviating too much from the pre-trained representation, or maybe it is the major driven force in the proposed object detection pre-training, as it enforces the region features to be close to an image-level pre-trained model like SwAW. Could you provide more ablation study on this part?
4. Have you tried other unsupervised pretraining models as initialization and embedding loss, e.g., BYOL, do you have any ideas or perspectives on this aspect?

**Limitations:**

The submission does not discuss the limitations, it is encouraged to include more discussion on the limitation and possible future works and improvements that could be done.

**Strengths And Weaknesses:**


**Strengths**
- The submission is well-written and easy to follow. The math definitions and theories are clear and concise.
- It is surprising that bounding box prediction merely supervised by the attention cues and self-supervising signal could achieve such superior object proposal results, which has not been done before.

**Weaknesses and parts to be improved**
- I listed some of my concerns in the questions, which I'm not very sure, the authors are encouraged to address my concerns and I would consider raising my score.
- The approach is simple but it seems the technical contributions are sparse, the major components of attention map generation and region proposal generation are lent from prior work.
- In section 2.4, the authors mention it _uses a pre-trained SWAV model to extract features_,  however, the details are not given. Specifically, is the regions cropped from the image and fed to the pre-trained SWAV backbone, or some feature cropping method is used, e.g., ROIalign.
- In the introduction, the authors mention they are inspired by DINO and referred to the attention maps in the figures, however in the method section, it actually uses the Normalized Cuts algorithm to obtain the attention map. I believe it is better to rewrite the part in the introduction to be more clear and indicate in the figure captions how the attention map is obtained.

**minor comments**
- In line 84, can then **be** generated
- In equation 3, FB is not defined
- In equation 8, B and R should be defined properly, I guess R should be replaced with other characters as it is already used in equation 1.
- In line 144, SWAV is not cited
- In line 212, Detreg should be DETReg
- In Tab.2 and Tab.3, citations are missing for the prior works.
- In Tab.4, I suggest adding the IOU metric for calculating Recall, which is more clear.

---

> ### Author Response · Authors · 2022-08-02
> **Response to Reviewer 34gJ (part i of iii)**
>
> >**Q1:** The approach is simple but it seems the technical contributions are sparse, the major components of attention map generation and region proposal generation are lent from prior work.
>
> **A1:** Thanks for your review. We do **NOT** lend region proposal generation from prior works. **INSTEAD**, our novelty is to generate proposals (object priors) from **both region priors and instance priors** (L118), where instance priors are generated by the matchings between attention maps and the predicted bounding boxes of the detector (L128), providing complementary foreground instances for supervision.
>
> Instance priors are very important for detector pretraining. Region priors alone provide only comparable performance  with DETReg as shown in the ablation study in Tab.5. Compared with using only region priors (**53.5 AP** on PASCAL VOC), adding instance priors (**55.4 AP** on PASCAL VOC) boosts JoinDet for **+1.9AP** , showing the importance of instance priors in bounding box generation.
>
> Furthermore, we propose two other contributions. First, we propose to **jointly** generate object priors and learn object detection which can provide progressively refined supervision. Second, we propose a **Box Smooth** method for box refinement, which stabilizes the pretraining during the evolvement of object prior generation.
>
> >**Q2:** In section 2.4, the authors mention it uses a pre-trained SWAV model to extract features, however, the details are not given. Specifically, is the regions cropped from the image and fed to the pre-trained SWAV backbone, or some feature cropping method is used, e.g., ROIalign.
>
> **A2:** Yes, we follow the process in UP-DETR[1] and DETReg[2], the regions cropped from the image are fed into the pretrained SwAV backbone.
>
> >**Q3:** In the introduction, the authors mention they are inspired by DINO and referred to the attention maps in the figures, however in the method section, it actually uses the Normalized Cuts algorithm to obtain the attention map. I believe it is better to rewrite the part in the introduction to be more clear and indicate in the figure captions how the attention map is obtained.
>
> **A3:** Thanks for your suggestion. We will revise this part in the introduction.
>
> >**Q4:** I actually have a question regarding the effectiveness of detection pre-training. For the Full-data finetuning experiment, the supervised training is compared as a baseline, however, would it be fair to add the pre-training time to the baseline? I guess when you extend the baseline training time, it would easily catch up with the detection pre-training methods, which would pose a concern on the effectiveness of detection pre-training for full-data finetuning.
>
> **A4:** Thanks for the comment. (1) Lots of previous papers [1,2,3] in this field use the supervised pretraining on ImageNet as a baseline pretraining method. In this paper, we follow this widely used baseline. (2) Following your suggestions, we extend the training epochs to 200 epochs for the supervised pretraining on PASCAL VOC and achieve 59.3 AP, which is still lower (**-5.1AP**) than our JoinDet (**64.4 AP**). On the relatively small dataset, PASCAL VOC, the supervised pretraining shows to be **over-fitting** with 200 epochs. We suggest that the performance gap between supervised pretraining and JoinDet verifies the effectiveness of detection pre-training.
>
> In the **special** case where pretraining data and fine-tuning are **exactly** the same in COCO, extending the training epochs for supervised pretraining to 100 epochs, we get a detection performance of 45.6 AP. Considering our JoinDet reaches **45.6 AP** using **only** 50 epochs of fine-tuning, we suggest that **JoinDet has reached the upper bound of deformable DETR on COCO** when we fine-tune the pretrained detector with 50 epochs.
>
> Still, we suggest that **fine-tuning using less data or fewer epochs** is a more common and valuable setting because it is closer to real needs and collecting larger-scale unlabeled pretraining data through the Internet is easy and feasible.
> | Finetune dataset | Methods    | Pretrain epochs on COCO without labels | Full-data Finetune epochs |  AP  |
> |------------------|------------|:-------------------------------:|:------------------:|:----:|
> | PASCAL VOC       | Supervised |                0                |         100        | 59.5 |
> | PASCAL VOC       | Supervised |                0                |         200        | 59.3 |
> | PASCAL VOC       | JoinDet    |                50               |         100        | **64.4** |
> | COCO             | Supervised |                0                |         50         | 44.5 |
> | COCO             | Supervised |                0                |         100        | 45.6 |
> | COCO             | JoinDet    |                50               |         50         | 45.6 |

---

> > ### Author Response · Authors · 2022-08-02
> > **Response to Reviewer 34gJ (part ii of iii)**
> >
> > >**Q5:** Although the work does not employ low-level unsupervised proposal methods like Selective Search, would it be more advantageous to incorporate these proposals which are generated from low-level cues? The intuition is that they may provide better or complementary coverage to some objects such as small ones, or others that have clear boundaries to the background.
> >
> > **A5:** **NO**. **Simply introducing low-level cues** leads to performance drops on downstream tasks. Following the ablation setting in Sec. 4, we add 30 Selective Search boxes with originally generated object priors as supervision during pretraining and find that additional selective search proposals lead to **-2.5AP** and **-0.5AP** performance drops with 10 epochs and 25 epochs finetuning on PASCAL VOC, respectively. We suggest that there are two reasons. (1) **Low-level cues lack semantic information** and will introduce lots of non-object-related supervision, which is harmful to detector pretraining when more accurate self-supervised information is presented. We refer the reviewers to see Fig.1 in the supplementary for visualizations. (2) JoinDet is already able to generate useful supervision for small objects and objects that have clear boundaries to the background. We use large-scale jittering mentioned in [4] to provide supervision for small objects.
> >
> > | Method                     | 10 epochs VOC | 25 epochs VOC |
> > |----------------------------|:-------------:|:-------------:|
> > | JoinDet                    |      49.0     |      55.3     |
> > | JoinDet + selective search |      46.5     |      54.8     |
> >
> > >**Q6:** I'm interested in how the pre-trained object feature embedding loss affects the method, have you tried removing the feature embedding loss? Do you have any perspectives, e.g., it is a regularization that avoids the feature representation deviating too much from the pre-trained representation, or maybe it is the major driven force in the proposed object detection pre-training, as it enforces the region features to be close to an image-level pre-trained model like SwAW. Could you provide more ablation study on this part?
> >
> > **A6:** The embedding loss encourages the detector to capture useful information for classification[1,2], and removing the embedding loss leads to **slight performance drops** on JoinDet, which is consistent with the ablation in Table 5 in UP-DETR[1]. Specifically, the table below shows that removing the embedding loss leads to a **-0.5AP** performance drop in early fine-tuning epochs. When fine-tuning 25 epochs, the drop decreases to **-0.1AP**. Similar to UP-DETR, when the backbone is frozen during pretraining, the embedding loss (named feature reconstruction loss in UP-DETR) only influences the finetuning performance in early epochs (shown in Fig.4 and Tab.5 in UP-DETR). As we do not claim the contribution on the loss design, we directly use this loss from UP-DETR[1] and DETReg[2].
> > | Method               | 10 epochs VOC | 25 epochs VOC |
> > |----------------------|:-------------:|:-------------:|
> > | JoinDet w/ emb loss  |      49.0     |      55.3     |
> > | JoinDet w/o emb loss |      48.5     |      55.2     |

---

> > > ### Author Response · Authors · 2022-08-02
> > > **Response to Reviewer 34gJ (part iii of iii)**
> > >
> > > >**Q7:** Have you tried other unsupervised pretraining models as initialization and embedding loss, e.g., BYOL, do you have any ideas or perspectives on this aspect?
> > >
> > > **A7:** Thanks for the comment. Using BYOL as the backbone brings **more improvement** to JoinDet. Following the ablation setting in Sec. 4, the experimental results below show that BYOL achieves better performance (**+0.6AP**) on full data VOC finetuning. In this paper,  as did in all previous works, we use SwAV to initialize the backbone for fair comparisons.
> > > | Method                  | 10 epochs VOC | 25 epochs VOC |
> > > |-------------------------|:-------------:|:-------------:|
> > > | JoinDet(Swav) - default |      49.0     |      55.3     |
> > > | JoinDet(BYOL)           |      49.8     |      55.9     |
> > >
> > > **Our perspectives:**
> > >
> > > (1) Usually, a powerful representation of image classification **benefits** downstream detection tasks. Concretely, on ImageNet classification, MoCo v2[5] shows +1.8 and +10.5 top-1 accuracy gains than SimCLR[6] and MoCo[7]. Accordingly, on COCO detection, MoCo v2[1] achieves +0.6 AP and +1.0 AP performance gains on COCO when compared with SimCLR[2] and MoCo[3], respectively (reported in [8]). Furthermore, on JoinDet, a powerful representation of image classification can produce better eigen attention maps and more accurate object priors to supervise the detector learning. Meanwhile, a better pretraining for backbone initialization can provide better target features in the embedding loss (L176) during pretraining[1].
> > >
> > > (2) However, the average accuracy on ImageNet may **NOT** be an absolute (though relatively good) metric to choose better unsupervised pretrainings for backbone initialization on the unsupervised object detection pretraining task. Specifically, SwAV[6] shows better performance (+1.0 top-1 accuracy) than BYOL, but shows lower performance (**-0.6 AP**) on JoinDet. We suggest that, as downstream datasets (COCO, Pascal VOC) contain mostly scene images, the accuracy of uncommon categories in ImageNet may not help the model represent objects in scene images. We will explore it in our future work.
> > >
> > > >**Q8:** Minor comments on writings.
> > >
> > > **A8:** We appreciate the reviewer's valuable writing comments and we will revise and update these parts in our future version.
> > >
> > > >**Q9:** In table.4, I suggest adding the IOU metric for calculating Recall, which is more clear.
> > >
> > > **A9:** Yes, we will add the IOU metric for calculating Recall. Compared with other methods, the higher Average Recall of JoinDet in Tab.4 shows that it detects more objects without supervision from the ground truth.
> > >
> > > >**Q10:** The submission does not discuss the limitations, it is encouraged to include more discussion on the limitation and possible future works and improvements that could be done.
> > >
> > > **A10:** We mention the limitations at the end of Sec.6 and we are willing to include more discussion in our future version.
> > >
> > > **Reference:**
> > >
> > > [1] Dai, Zhigang, et al. "Up-detr: Unsupervised pre-training for object detection with transformers." Proceedings of the IEEE/CVF conference on computer vision and pattern recognition. 2021.
> > >
> > > [2] Bar, Amir, et al. "Detreg: Unsupervised pretraining with region priors for object detection." Proceedings of the IEEE/CVF Conference on Computer Vision and Pattern Recognition. 2022.
> > >
> > > [3] Zhong, Yuanyi, et al. "Dap: Detection-aware pre-training with weak supervision." Proceedings of the IEEE/CVF Conference on Computer Vision and Pattern Recognition. 2021.
> > >
> > > [4] Ghiasi, Golnaz, et al. "Simple copy-paste is a strong data augmentation method for instance segmentation." Proceedings of the IEEE/CVF Conference on Computer Vision and Pattern Recognition. 2021.
> > >
> > > [5] Chen, Xinlei, et al. "Improved baselines with momentum contrastive learning." arXiv preprint arXiv:2003.04297 (2020).
> > >
> > > [6] Chen, Ting, et al. "A simple framework for contrastive learning of visual representations." International conference on machine learning. PMLR, 2020.
> > >
> > > [7] He, Kaiming, et al. "Momentum contrast for unsupervised visual representation learning." Proceedings of the IEEE/CVF conference on computer vision and pattern recognition. 2020.
> > >
> > > [8] Xie, Zhenda, et al. "Propagate yourself: Exploring pixel-level consistency for unsupervised visual representation learning." Proceedings of the IEEE/CVF Conference on Computer Vision and Pattern Recognition. 2021.

---

### Official Review · Reviewer_EWLn · 2022-07-11

**Rating:** 5
**Confidence:** 4
**Soundness:** 3 good
**Presentation:** 3 good
**Contribution:** 2 fair

**Summary:**

This paper presents an unsupervised pre-training strategy for DETR-Like object detectors. The core idea is to generate object priors from the encoder to guide the self-supervised pre-training.




**Questions:**

See weakness.

**Limitations:**

Yes. The authors mentioned "there still exists a large gap from supervised training on class agnostic object proposal evaluation, which calls for further studies."

**Strengths And Weaknesses:**

Strengths:
* Using the encoder itself to generate object priors is a good idea.
* The proposed method outperforms other methods designed for pre-training DETR-like object detectors.

Weakness:
* Missing important comparisons: For a rough categorization, the previous contrastive pre-training methods focused on (1)  **image classification** (e.g., Mocov2 and SimCLR) (2) **general dense prediction tasks**[1, 2, 3, 4] (such as object detection and segmentation) (3) **detr-like object detectors**. This paper belongs to (3), and only compared previous works of (1) and (3). But the detailed comparisons to the (2) are missing. It would be much better if the authors can provide deep discussions.
* The advantages of pre-training is the generalization to various downstream tasks. I wonder if the representation learned in this method can be also used for other tasks such as segmentation.

[1] Propagate Yourself: Exploring Pixel-Level Consistency for Unsupervised Visual Representation Learning. CVPR 2021.
[2] Dense Contrastive Learning for Self-Supervised Visual Pre-Training. CVPR 2021.
[3] Region Similarity Representation Learning. ICCV 2021.
[4] Deeply Unsupervised Patch Re-Identification for Pre-training Object Detectors. TPAMI 2022.

---

> ### Author Response · Authors · 2022-08-02
> **Response to Reviewer EWLn**
>
> >**Q1:** Missing important comparisons: For a rough categorization, the previous contrastive pre-training methods focused on (1) image classification (e.g., Mocov2 and SimCLR) (2) general dense prediction tasks[1, 2, 3, 4] (such as object detection and segmentation) (3) detr-like object detectors. This paper belongs to (3), and only compared previous works of (1) and (3). But the detailed comparisons to the (2) are missing. It would be much better if the authors can provide deep discussions. Compare with general dense prediction tasks.
>
> **A1:** Thanks for your suggestion. We will add the results of the papers you mentioned in type (2) in our paper. The difference between type (2) and type (3): First, type (2) **only pretrains the backbone** with dense prediction tasks, while type (3) directly pretrains **all the detection components**. Type (2) neglects the detector heads, when transferred to downstream detection tasks, the transformer part in deformable DETR is initialized from scratch and does not benefit from pretraining. Second, type (2) focuses on dense contrastive learning which can learn more fine-grained features but can **NOT** empower the model to learn the location of objects. Instead, type (3) pretrains the whole detector which targets on **spatial localization learning** which is important for downstream detection tasks.
>
> The experimental results below show that type (2) methods have lower performance (about **-4 AP** on low-data regime COCO detection) than our JoinDet. Specifically, we evaluate the performance of ReSim[3] (which is shown in Table 1 of the paper) and PixPro[1] (added in the rebuttal) on the low-data regime COCO fine-tuning tasks. We only choose these two methods as they achieve relatively better performance than DenseCL[2] (reported in [6]) and we can not find released checkpoints for DUPR[4]. Compared with PixPro, JoinDet achieves **+4.05 AP** and **+4.63 AP** performance gains when fine-tuning on 1% and 10% COCO data, respectively.
> | Method    |   Pretrain  |  COCO 1%  |  COCO 10%  |
> |-----------|:-----------:|:---------:|:----------:|
> | ReSim[3]  |   Backbone  | 11.07±0.4 |  26.56±0.3 |
> | PixPro[1] |   Backbone  | 11.84±0.3 |  26.24±0.2 |
> | DETReg[5] | Whole model | 14.58±0.3 | 29.12±0.2  |
> | JoinDet   | Whole model | 15.89±0.2 |  30.87±0.1 |
>
> >**Q2:** The advantage of pre-training is the generalization to various downstream tasks. I wonder if the representation learned in this method can be also used for other tasks such as segmentation.
>
> **A2:** **YES**, our method can be used for segmentation. There are two directions to extend our method for segmentation tasks. (1) After pretraining, given foreground object bounding boxes learned by JoinDet, the segmentation masks in the bounding boxes can be easily achieved by adding a mask branch like that in Mask-RCNN and DETR. (2) During pretraining, as the eigen attention maps and object priors are progressively refined, highlighted foreground regions in the eigen attention maps bounded by object priors can be treated as the supervision for segmentation. We will explore this extended version in our future works.
>
> **Reference:**
>
> [1] Xie, Zhenda, et al. "Propagate yourself: Exploring pixel-level consistency for unsupervised visual representation learning." Proceedings of the IEEE/CVF Conference on Computer Vision and Pattern Recognition. 2021.
>
> [2] Wang, Xinlong, et al. "Dense contrastive learning for self-supervised visual pre-training." Proceedings of the IEEE/CVF Conference on Computer Vision and Pattern Recognition. 2021.
>
> [3] Xiao, Tete, et al. "Region similarity representation learning." Proceedings of the IEEE/CVF International Conference on Computer Vision. 2021.
>
> [4] Ding, Jian, et al. "Deeply Unsupervised Patch Re-Identification for Pre-training Object Detectors." IEEE Transactions on Pattern Analysis and Machine Intelligence 2022.
>
> [5] Bar, Amir, et al. "Detreg: Unsupervised pretraining with region priors for object detection." Proceedings of the IEEE/CVF Conference on Computer Vision and Pattern Recognition. 2022.
>
> [6] Wei, Fangyun, et al. "Aligning pretraining for detection via object-level contrastive learning." Advances in Neural Information Processing Systems. 2021.

---

> > ### Comment · Reviewer_EWLn · 2022-08-08
> > **Responses**
> >
> > Thank you for your responses. Generaly, I think you addressed most of my concerns. But I still have some questions.
> > For A1, what is the performance of [6] on the low-data regime COCO fine-tuning tasks? And do you have comparisons with type (2) on normal COCO setting (not low-data regime)?

---

> > > ### Author Response · Authors · 2022-08-09
> > > **Responses to Reviewer EWLn**
> > >
> > > >Q: For A1, what is the performance of [6] on the low-data regime COCO fine-tuning tasks?
> > >
> > > A: JoinDet still achieves **considerable performance gains** compared with SoCo[6]. Specifically, we download the pretrained model from the official repo, and then finetune the whole detection model with the pretrained backbone. Experimental results show that our JoinDet improves SoCo[6] by **+4.40 AP** and **+4.46 AP** on 1% and 10% COCO.
> > >
> > > We would suggest SoCo[6] belongs to type (2), and type (2) pretraining methods are generally worse than type (3) pretraining methods, e.g., our JoinDet (as shown in Tab.1 of our paper and Tab.5 of DETReg) as discussed in **A1** of the **Responses to your review (Response to Reviewer EWLn)**, please refer to it for detailed explanations.
> > > | Method  | Pretrain    |    COCO 1%    |    COCO 10%   |
> > > |---------|-------------|:-------------:|:-------------:|
> > > | SoCo[6] | Backbone    |     11.49     |     26.41     |
> > > | JoinDet | Whole model | **15.89** (+4.40) | **30.87** (+4.46) |
> > >
> > > >Q: And do you have comparisons with type (2) on normal COCO setting (not low-data regime)?
> > >
> > > A: Our JoinDet generally has significant performance gain (about **+8 AP50**)  compared with type (2) on normal COCO setting. Specifically, we evaluate the performance of JoinDet (see Fig. 4 of our paper) as well as ReSim and PixPro (see the following table). For ReSim and PixPro, we download the pretrained model from the official GitHub repo, and then fine-tune the detector with the pretrained backbone on normal COCO setting with 10 epochs due to the time limit. The performance drop of type (2) compared with our JoinDet results from neglecting the detector head during pretraining. Detailed discussion about the difference between type (2) and type (3) can be also found in **A1** of the **Responses to your review (Response to Reviewer EWLn)**.
> > > | Method    | Pretrain    | COCO AP50 |
> > > |-----------|-------------|:---------:|
> > > | ReSim[1]  | Backbone    |    44.2   |
> > > | PixPro[2] | Backbone    |    44.8   |
> > > | JoinDet   | Whole model |    **53.0**   |

---

### Official Review · Reviewer_oHTr · 2022-07-13

**Rating:** 5
**Confidence:** 4
**Soundness:** 3 good
**Presentation:** 3 good
**Contribution:** 2 fair

**Summary:**

This paper focuses on unsupervised object detection pre-training using object priors. Specifically, two kinds of object priors are used in this paper. The first one is region prior which is generated from the encoder attention maps. The second one is instance prior which is obtained by selecting high-quality outputs from the detector decoder. Experiments on COCO and PASCAL VOC show that the proposed approach obtains better results than the previous unsupervised object detection pre-training approaches when only limited number of annotated data is available.

**Questions:**

From Table 3, when using the full COCO training set, only marginal improvement (0.1%) is obtained by the proposed approach comparing to the previous state-of-the-art object prior based pre-training approach DETReg. Is there any explanation about why this happens and any thought on how to improve it?

**Limitations:**

Only marginal improvement (0.1%) is obtained by the proposed approach comparing to the previous state-of-the-art object prior based pre-training approach DETReg.

**Strengths And Weaknesses:**

### Strengths
- The proposed approach is interesting.
- Promising results are obtained when limited number of annotated data is available.

### Weaknesses
- Novelty of region prior. There is already published work which uses attention maps to generate object bounding boxes to pre-train object detector [a]. The authors should discuss and compare with this paper.
- Marginal improvement when more annotated data is available. From Table 3, when using the full COCO training set, only marginal improvement (0.1%) is obtained by the proposed approach comparing to the previous state-of-the-art object prior based pre-training approach DETReg.

[a] DAP: Detection-Aware Pre-training with Weak Supervision

---

> ### Author Response · Authors · 2022-08-02
> **Response to Reviewer oHTr (part i of ii)**
>
> >**Q1:** Novelty of region prior. There is already published work which uses attention maps to generate object bounding boxes to pre-train object detector [1]. The authors should discuss and compare with this paper.
>
> **A1:** (1) Thanks for your comments. We would like to clarify that the region prior generation alone by the attention map is **NOT** our claimed novelty. Instead, our novelty is to **generate bounding boxes from both region priors and instance priors** (Line118), where instance priors that are generated by the matchings between attention maps and the predicted bounding boxes of the detector are new to the best of our knowledge.
>
> **Instance priors can provide complementary information for region priors** and are very important for detector pretraining. Specifically, region priors only provide comparable performance with DETReg as shown in the ablation study in Tab.5. Compared with using only region priors (**53.5 AP** on PASCAL VOC), adding instance priors (**55.4 AP** on PASCAL VOC) boosts JoinDet for **+1.9 AP**, showing the importance of instance prior in bounding box generation. The joint generation of region prior and instance prior corresponds to our claimed **contribution #1** in L59-61. Furthermore, our paper has deserved the other two contributions:
>
> **Contribution #2** (L61-63): our design can generate object priors (including regions priors and instance priors) and learn object detection **synchronously**, which brings **mutual evolvement** to the object prior generation and detector optimization.
>
> **Contribution #3** (L63-64): a **Box Smooth Module** is proposed to stabilize the pretraining during the refinement of generated object priors.
>
> (2) Generally, our novelty is **independent** of DAP[1], but we appreciate the reminding of the reviewer and will add it to the related works. Their differences can be summarized as three folds. First, JoinDet is totally **unsupervised** while DAP **<u><u>relies on image labels</u></u>** to generate Class Activation Maps (CAMs), which can not work without ground-truth class labels. Second, JoinDet further generates instance priors via matching the attention maps with the predicted bounding boxes and can optimize object priors and detector learning synchronously, while DAP does **NOT** generate instance priors. Third, JoinDet designs a Box Smooth Module while DAP does **NOT**.
>
> These differences bring three advantages. First, as an unsupervised method, JoinDet can **be easily adapted to** newly collected, unlabelled datasets. Second, JoinDet progressively refines boxes to provide more and more accurate supervision during pretraining. Third, generated instance priors in JoinDet divide foreground regions into foreground instances, which provides more **object-related boxes** for supervision and is more suitable for scene image datasets.

---

> > ### Author Response · Authors · 2022-08-02
> > **Response to Reviewer oHTr (part ii of ii)**
> >
> > >**Q2:** Marginal improvement when more annotated data is available. From Table 3, when using the full COCO training set, only marginal improvement (0.1%) is obtained by the proposed approach comparing to the previous state-of-the-art object prior based pre-training approach DETReg. (1) Is there any explanation about why this happens and (2) any thought on how to improve it?
> >
> > **A2:** **Explanation:** Experiments in Tab.3 are conducted in a **special** setting, where the pretraining data and fine-tuning data are **exactly** the same and the pretrained models are finetuned by **sufficiently long epochs** (i.e., deformable DETR almost converges with 50 epochs training [2]). More importantly, when we extend the fine-tuning epochs for supervised pretraining to 100 epochs, we get a detection performance of 45.6 AP. Considering our JoinDet reaches **45.6 AP** using **only** 50 epochs of fine-tuning, we suggest that **JoinDet has reached the upper bound of deformable DETR on COCO** when we fine-tune the pretrained model with 50 epochs.
> > | Finetune dataset |   Methods  | Pretrain epochs on COCO without labels | Full-data Finetune epochs |  AP  |
> > |:----------------:|:----------:|:-------------------------------:|:------------------:|:----:|
> > |       COCO       | Supervised |                0                |         50         | 44.5 |
> > |       COCO       | Supervised |                0                |         100        | 45.6 |
> > |       COCO       |   JoinDet  |                50               |         50         | 45.6 |
> >
> > We consider such a specific setting unsuitable for evaluating the performance of pretraining methods because sufficiently long fine-tuning epochs lead to less disparity among different pretrainings. We provide these results just for making the evaluation more comprehensive following existing works.
> >
> > We would like to further highlight that **fine-tuning with less data or fewer epochs** is a more common and valuable evaluation setting, with which our JoinDet achieves considerable improvements compared with DETReg. Specifically, on full-data PASCAL VOC fine-tuning, JoinDet improves DETReg by **+1.0 AP**. When 1% COCO data are used for fine-tuning, JoinDet shows a **+1.31 AP** performance gain on DETReg. When fine-tuning with fewer (10) epochs on COCO, DETReg achieves a **+3.3 AP50** performance gain when compared with DETReg (shown in Fig. 4).
> >
> > **Methods to improve the finetuning results on full coco data:** we would suggest using larger unlabeled pretraining datasets, such as COCO+ and OpenImages, which are easy to obtain on the Internet and bring out the full potential of JoinDet.
> >
> > **Reference**
> >
> > [1] Zhong, Yuanyi, et al. "Dap: Detection-aware pre-training with weak supervision." Proceedings of the IEEE/CVF Conference on Computer Vision and Pattern Recognition. 2021.
> >
> > [2] Zhu, Xizhou, et al. "Deformable DETR: Deformable Transformers for End-to-End Object Detection." International Conference on Learning Representations. 2020.

---

### Author Response · Authors · 2022-08-08
**We would be happy to address any follow-up questions**

We sincerely thank you for the reviews and comments. We have provided corresponding responses and results, which we believe have covered your concerns. We hope to further discuss with you whether or not your concerns have been addressed. Please let us know if you still have any unclear parts of our work. We would be happy to address any follow-up questions.

Best,

Authors of Paper 2341

---

### Author Response · Authors · 2022-08-09
**A Gentle Reminder**

Thank you for your time and efforts in reviewing our paper!

We kindly remind you that the discussion period will end in half a day, and thus we just wonder whether we could have the last chance to address your further concerns or questions (if you have any). We are sincerely glad to improve our paper under your suggestions!

Best,

Authors of Paper 2341

---

### Meta-Review · Area_Chair_jMnt · 2022-08-27

**Recommendation:** Accept
**Confidence:** Certain

**Metareview:**

The paper received mixed reviews. Three reviewers rated borderline accept and one reviewer rated borderline reject. The authors provided detailed responses to the raised concerns/questions and supported their responses with additional ablation study, experimental result on new dataset (e.g., VOC).

For reviewer fpzy (who gave borderline reject), the requested additional analysis have been provided by the authors. The major remaining issue is "The improvement over DETReg is somewhat limited". The results presented in the paper did show consistent improvement over DETReg on three settings with at least 1 mAP improvement.

After reading the reviews and the responses, while there are no enthusiastic supports from the reviewers, the AC does not find sufficient ground to reject the paper. This paper introduces new ideas for unsupervised object detection pretraining and show consistent improvement over the baselines over three evaluation settings. The AC believes that this work would benefit the community and thus recommends to accept.

**Award:**

No

---

### Decision · Program_Chairs · 2022-09-14

Accept